# Oncogenic and teratogenic effects of *Trp53*[Y217C], an inflammation-prone mouse model of the human hotspot mutant *TP53*[Y220C]

Sara Jaber[1†], Eliana Eldawra[1†], Jeanne Rakotopare[1†], Iva Simeonova[2], Vincent Lejour[1], Marc Gabriel[3], Tatiana Cañeque[4], Vitalina Volochtchouk[1], Monika Licaj[1], Anne Fajac[1], Raphaël Rodriguez[4], Antonin Morillon[3], Boris Bardot[1,5]*, Franck Toledo[1,6]*

[1]Genetics of Tumor Suppression, Institut Curie, CNRS UMR3244, Sorbonne University, PSL University, Paris, France; [2]Chromatin Dynamics, Institut Curie, CNRS UMR3664, Sorbonne University, PSL University, Paris, France; [3]Non Coding RNA, Epigenetic and Genome Fluidity, Institut Curie, CNRS UMR3244, Sorbonne University, PSL University, Paris, France; [4]Chemical Biology, Institut Curie, CNRS UMR3666, INSERM U1143, PSL University, Paris, France; [5]Signaling and Neural Crest Development, Institut Curie, CNRS UMR3347, INSERM U1021, Université Paris-Saclay, PSL University, Orsay, France; [6]Hematopoietic and Leukemic Development, Centre de Recherche Saint-Antoine, INSERM UMRS938, Sorbonne University, Paris, France

*For correspondence:
boris.bardot@curie.fr (BB);
franck.toledo@sorbonne-universite.fr (FT)

†These authors contributed equally to this work

Competing interest: The authors declare that no competing interests exist.

## eLife Assessment

This work is of **fundamental** significance and has an **exceptional** level of evidence for the role of a mutant p53 in regulation of tumorigenesis using an in vivo mouse model. The study is well-conducted and will be of interest to a broad audience including those interested in p53, transcription factors and cancer biology.

**Abstract** Missense 'hotspot' mutations localized in six p53 codons account for 20% of *TP53* mutations in human cancers. Hotspot p53 mutants have lost the tumor suppressive functions of the wildtype protein, but whether and how they may gain additional functions promoting tumorigenesis remain controversial. Here, we generated *Trp53*[Y217C], a mouse model of the human hotspot mutant *TP53*[Y220C]. DNA damage responses were lost in *Trp53*[Y217C/Y217C] (*Trp53*[YC/YC]) cells, and *Trp53*[YC/YC] fibroblasts exhibited increased chromosome instability compared to *Trp53*[-/-] cells. Furthermore, *Trp53*[YC/YC] male mice died earlier than *Trp53*[-/-] males, with more aggressive thymic lymphomas. This correlated with an increased expression of inflammation-related genes in *Trp53*[YC/YC] thymic cells compared to *Trp53*[-/-] cells. Surprisingly, we recovered only one *Trp53*[YC/YC] female for 22 *Trp53*[YC/YC] males at weaning, a skewed distribution explained by a high frequency of *Trp53*[YC/YC] female embryos with exencephaly and the death of most *Trp53*[YC/YC] female neonates. Strikingly, however, when we treated pregnant females with the anti-inflammatory drug supformin (LCC-12), we observed a fivefold increase in the proportion of viable *Trp53*[YC/YC] weaned females in their progeny. Together, these data suggest that the p53[Y217C] mutation not only abrogates wildtype p53 functions but also promotes inflammation, with oncogenic effects in males and teratogenic effects in females.

## Introduction

Somatic alterations in the *TP53* gene, encoding the tumor suppressor p53, are the most common events in human tumors (*Hollstein et al., 1991*). The p53 protein is a stress sensor stabilized and activated in response to potentially oncogenic insults. In its tetrameric active form, wildtype (WT) p53 can trigger a transcriptional program to induce various responses including cell cycle arrest, senescence, apoptosis, or metabolic changes (*Beckerman and Prives, 2010*). In human cancers, about 75% of all *TP53* alterations are missense mutations, and the eight most frequent ('hotspot') missense mutations (R175H, Y220C, G245S, R248Q, R248W, R273H, R273C, and R282W) affect six residues localized in the p53 DNA-binding domain (DBD) and account for 20% of all *TP53* mutations (*Hainaut and Pfeifer, 2016*). The fact that most *TP53* mutations are missense substitutions suggested that cells expressing mutant p53 might have a selective advantage over cells lacking p53, and evidence for this was first gained by expressing various p53 mutants in p53-null cells and observing enhanced tumorigenic potential in nude mice or increased plating efficiency in agar cell culture (*Dittmer et al., 1993*). Mouse models expressing hotspot p53 mutants next helped to define potential mechanisms accounting for accelerated tumorigenesis. First, a dominant-negative effect (DNE) may be observed in heterozygotes, if the mutant p53 inhibits the WT p53 protein in hetero-tetramers (*Gencel-Augusto and Lozano, 2020*). Evidence that this mechanism accounts for accelerated tumorigenesis was notably reported in human leukemia (*Boettcher et al., 2019*). A second mechanism is a gain of function (GOF), i.e., the acquisition by mutant p53 of new oncogenic properties (*Amelio and Melino, 2020*; *Stein et al., 2020*). Although the DNA sequence specificity of p53 mutants is impaired, leading to a loss of WT functions (loss of function [LOF]), many p53 mutant proteins are stabilized in the cell and might engage in aberrant interactions with other transcription factors, chromatin-modifying complexes, or DNA helicase subunits, leading to the acquisition of GOF phenotypes (*Kim and Deppert, 2007*; *Pfister and Prives, 2017*; *Zhao et al., 2024*). However, a few recent studies challenged the pathological importance of mutant p53 GOF, or at least its relevance in anti-cancer therapeutic strategies (*Aubrey et al., 2018*; *Boettcher et al., 2019*; *Wang et al., 2024*). An alternative hypothesis emerged, postulating that the tumorigenic properties of a p53 mutant might result from a separation of function (SOF), if the mutant retains pro-proliferative or pro-survival functions of WT p53 while losing its tumor suppressive activities (*Kennedy and Lowe, 2022*). The concept of SOF was first proposed for *TP53* exon 6 truncating mutations, which mimic a naturally occurring alternative p53 splice variant (*Shirole et al., 2016*), but may apply to some p53 missense mutations, e.g., p53$^{R248W}$ (*Humpton et al., 2018*). Importantly, the analysis of various mutant p53 alleles in vivo appears crucial for a better understanding of their contribution to tumorigenesis, because GOF/SOF phenotypes may vary depending on the mutated residue, its specific mutation, the cellular context, or genetic background (*Dibra et al., 2024*; *Hanel et al., 2013*; *Kadosh et al., 2020*; *Kim and Lozano, 2018*; *Kotler et al., 2018*; *McCann et al., 2022*; *Xiong et al., 2022*).

In human cancers, the *TP53*$^{Y220C}$ mutation is the most frequent missense mutation outside of the DNA-binding surface of p53 (*Hainaut and Pfeifer, 2016*). The somatic *TP53*$^{Y220C}$ mutation accounts for over 100,000 new cancer cases per year worldwide, including solid tumors and myeloid neoplasms, and a germline *TP53*$^{Y220C}$ mutation was reported in 15 families with the Li-Fraumeni syndrome of cancer predisposition (*Bouaoun et al., 2016*; *Gener-Ricos et al., 2024*). The mutation causes a structural change in the DBD localized at the periphery of the β-sandwich region of the protein, far from the surface contact of DNA. The mutation from a tyrosine to a cysteine markedly lowers the thermodynamic stability of the DBD, leading to a largely unfolded and inactive protein at body temperature (*Joerger et al., 2006*), and a molecule designed to bind p53$^{Y220C}$ and restore WT protein conformation appears as a promising anti-cancer drug (*Dumbrava et al., 2022*). Analyses of the impact of p53$^{Y220C}$ in cancer cell lines led to conflicting results. On one hand, overexpression of p53$^{Y220C}$ in a p53-null cell line increased its capacity for migration or invasion (*Zhou et al., 2022*), and a decreased expression of p53$^{Y220C}$, caused by RNA interference or various chemical compounds, promoted cell death or decreased the migratory or invasive capacities of cells (*Tseng et al., 2022*; *Vikhanskaya et al., 2007*; *Zhou et al., 2022*). On the other hand, the removal by CRISPR/Cas9 of p53 mutants with reported GOF - including p53$^{Y220C}$ - in diverse cancer cell lines had no significant impact on cell survival or proliferation, and the cell death caused by RNA interference against p53$^{Y220C}$ might result from nonspecific toxic effects (*Wang et al., 2024*). Here, to gain a better understanding of the impact of the p53$^{Y220C}$ mutation in vivo, we generated a mouse model with a targeted *Trp53*$^{Y217C}$ mutation - the mouse

equivalent to human *TP53*[Y220C] (*Figure 1—figure supplement 1*) - and analyzed animals and cells carrying one or two copies of the mutant allele.

## Results

### Targeting of a p53[Y217C] mutation in the mouse

We used homologous recombination in 129/SvJ embryonic stem (ES) cells to target the p53[Y217C] mutation at the mouse *Trp53* locus. Mice expressing p53[Y217C] conditionally were generated by using a targeting vector containing transcriptional stops flanked by LoxP sites (LSL) upstream of coding sequences, and the p53[Y217C] mutation in exon 6 (*Figure 1A–D*). *Trp53*[+/LSL-Y217C] mice were then crossed with mice carrying the PGK-Cre transgene (*Lallemand et al., 1998*) to excise the LSL cassette and obtain *Trp53*[+/Y217C] mice, expressing the mutant protein constitutively. After two backcrosses over a C57BL/6J background, we prepared mouse embryonic fibroblasts (MEFs) from intercrosses of *Trp53*[+/Y217C] mice (*Figure 1E*). We extracted RNAs from *Trp53*[Y217C/Y217C] (*Trp53*[YC/YC]) MEFs then sequenced p53 mRNAs to verify that the p53[Y217C] coding sequence was identical to the wildtype p53 (p53[WT]) sequence, except for the desired missense mutation and a silent mutation introduced to facilitate mouse genotyping (*Figure 1F*). The quantification of p53 mRNA levels from WT and *Trp53*[YC/YC] MEFs next revealed similar transcription from both alleles (*Figure 1G*).

### Deficient p53-dependent stress responses in *Trp53*[YC/YC] cells

We used western blots to analyze protein extracts from *Trp53*[+/+], *Trp53*[+/-], *Trp53*[+/YC], *Trp53*[YC/YC], and *Trp53*[-/-] MEFs, unstressed or treated with Nutlin, an antagonist of Mdm2, the E3 ubiquitin ligase for p53 (*Vassilev et al., 2004*). Results indicated a high increase in p53 abundance in untreated and Nutlin-treated *Trp53*[YC/YC] MEFs and a moderate increase in untreated and Nutlin-treated *Trp53*[+/YC] MEFs, compared to WT cells (*Figure 2A*). Protein levels for p21 and Mdm2, the products of two classical p53 target genes, appeared similar in *Trp53*[+/+] and *Trp53*[+/YC] MEFs, and were undetectable or markedly decreased in *Trp53*[YC/YC] and *Trp53*[-/-] MEFs (*Figure 2A*). Accordingly, p53[Y217C] appeared unable to transactivate the *Cdkn1a* (alias *p21*) and *Mdm2* genes or to bind their promoters (*Figure 2B*). The fractionation of cells before or after treatment with doxorubicin, a clastogenic drug, indicated that p53[WT] accumulated in the nucleoplasm and chromatin fractions in response to DNA damage, whereas p53[Y217C] appeared mostly cytoplasmic in both untreated and doxorubicin-treated cells, and undetectable or barely detectable in chromatin fractions (*Figure 2C*). Furthermore, when Nutlin-treated MEFs were analyzed by immunofluorescence, p53[WT] was only detected in nuclei, whereas p53[Y217C] could be observed in nuclear and cytoplasmic compartments (*Figure 2D*). We next analyzed two well-known p53-mediated responses to DNA damage, i.e., cell cycle arrest in MEFs and apoptosis in thymocytes (*Kastan et al., 1992*; *Lowe et al., 1993*). WT MEFs exposed to increasing doses of γ-irradiation (3 or 12 Gy) exhibited significant increases in G1/S ratios, whereas G1/S ratios were similar before or after irradiation in *Trp53*[YC/YC] and *Trp53*[-/-] MEFs (*Figure 2E*, *Figure 2—figure supplement 1*). Furthermore, thymocytes recovered from irradiated WT mice underwent a massive apoptotic response, with an almost threefold increase in apoptotic cells after a 10 Gy irradiation. By contrast, no increase in apoptotic cells was observable upon irradiation in the thymi from *Trp53*[YC/YC] mice, and apoptotic thymocytes were equally rare in irradiated *Trp53*[YC/YC] and *Trp53*[-/-] mice (*Figure 2F*, *Figure 2—figure supplement 2*). Together, these results indicated that the p53[Y217C] mutation altered the abundance and intracellular distribution of the p53 protein, associated with a decrease in DNA-bound protein, and that p53[Y217C] had completely lost the ability to induce cell cycle arrest and apoptosis upon DNA damage.

### Increased chromosomal instability in *Trp53*[YC/YC] cells

We next searched for evidence, at the cellular level, of potential DNE or GOF for the p53[Y217C] mutant. Little if any difference in the transactivation of canonical p53 target genes was observed between WT, *Trp53*[+/YC], and *Trp53*[+/-] cells treated with Nutlin or Doxorubicin (*Figure 2—figure supplement 3A*), and *Trp53*[+/YC] and *Trp53*[+/-] cells exhibited similar cell cycle arrest or apoptotic responses to γ-irradiation (*Figure 2—figure supplement 3B and C*), suggesting little or no DNE in response to various stresses.

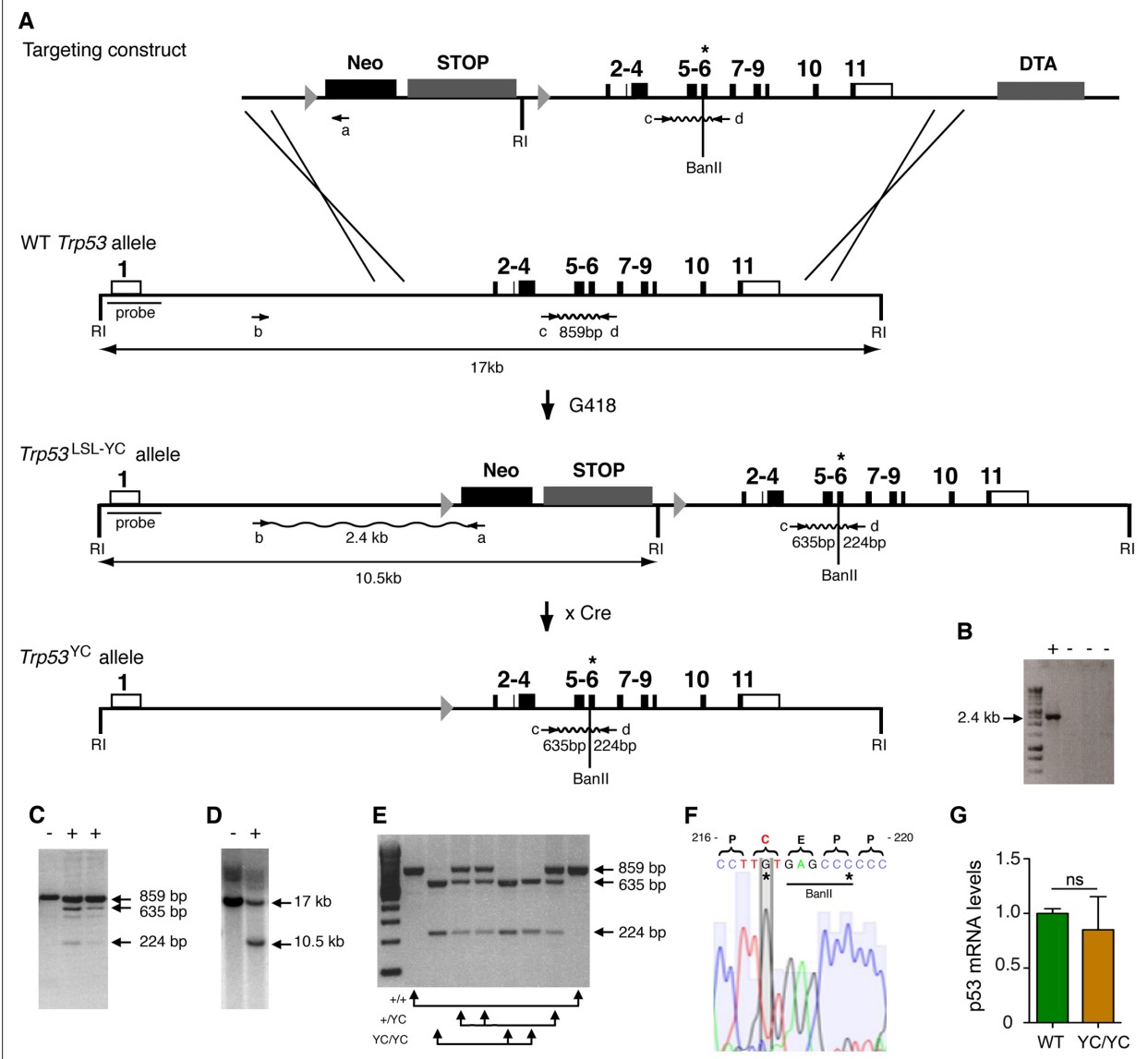

**Figure 1.** Targeting the Y217C missense mutation at the mouse *Trp53* locus. (**A**) Targeting strategy. The wildtype (WT) *Trp53* gene is within a 17-kb-long EcoRI (RI) fragment (black boxes are for coding sequences and white boxes for UTRs). The targeting construct contains: (1) a 1.5-kb-long 5′ homology region; (2) a Lox-Stop-Lox (LSL) cassette with a *neomycin* selection gene (Neo), four transcriptional stops (STOP) and an EcoRI site, flanked by LoxP sites (arrowheads); (3) p53 coding exons, including the Y217C (YC) missense mutation in exon 6 (asterisk) and an additional BanII site; (4) a 2.8-kb-long 3′ homology region; and (5) the diphteria α-toxin (DTA) gene for targeting enrichment. Proper recombinants with a *Trp53*^LSL-Y217C allele, resulting from the described crossing-overs, were G418 resistant. They were identified by a 2.4-kb-long band after PCR with primers a and b, and confirmed by bands of 635 and 224 bp after PCR with primers c and d and BanII digestion. They were also verified by Southern blot with the indicated probe as containing a 10.5 kb EcoRI band. Two recombinant ES clones were injected into blastocysts to generate chimeras, and germline transmission was verified by genotyping with primers c and d and BanII digestion. Excision of the LSL cassette was performed in vivo, by breeding *Trp53*^+/LSL-Y217C male mice with females carrying the PGK-*Cre* transgene, to obtain mice with a *Trp53*^Y217C allele. (**B–D**) Screening of recombinant ES clones (+) by PCR with primers a and b (**B**); PCR with primers c and d then BanII digestion (**C**); Southern blot (**D**). (**E**) Genotyping of mouse embryonic fibroblasts (MEFs) from an intercross of *Trp53*^+/Y217C mice, by PCR with primers c and d and BanII digestion. (**F**) *Trp53*^Y217C sequence around codon 217. The introduced Y217C missense mutation and the silent mutation creating an additional BanII restriction site are highlighted (asterisks). (**G**) WT and *Trp53*^Y217C/Y217C (YC/YC) MEFs express similar p53 mRNA levels. Total RNA was extracted, then p53 mRNAs were quantified by real-time qPCR, normalized to control mRNAs and the amount in WT cells was assigned the value of 1. Means + SEM (n=3) are shown. Primer sequences are listed in ***Supplementary file 5***.

The online version of this article includes the following source data and figure supplement(s) for figure 1:

**Source data 1.** Labeled files for gels and blots in ***Figure 1B, C, D, and E***.

**Source data 2.** Raw and unedited gels and blots for ***Figure 1B, C, D, and E***.

**Figure supplement 1.** Protein sequence alignment showing homology between mouse p53 Tyrosine 217 and human p53 Tyrosine 220.

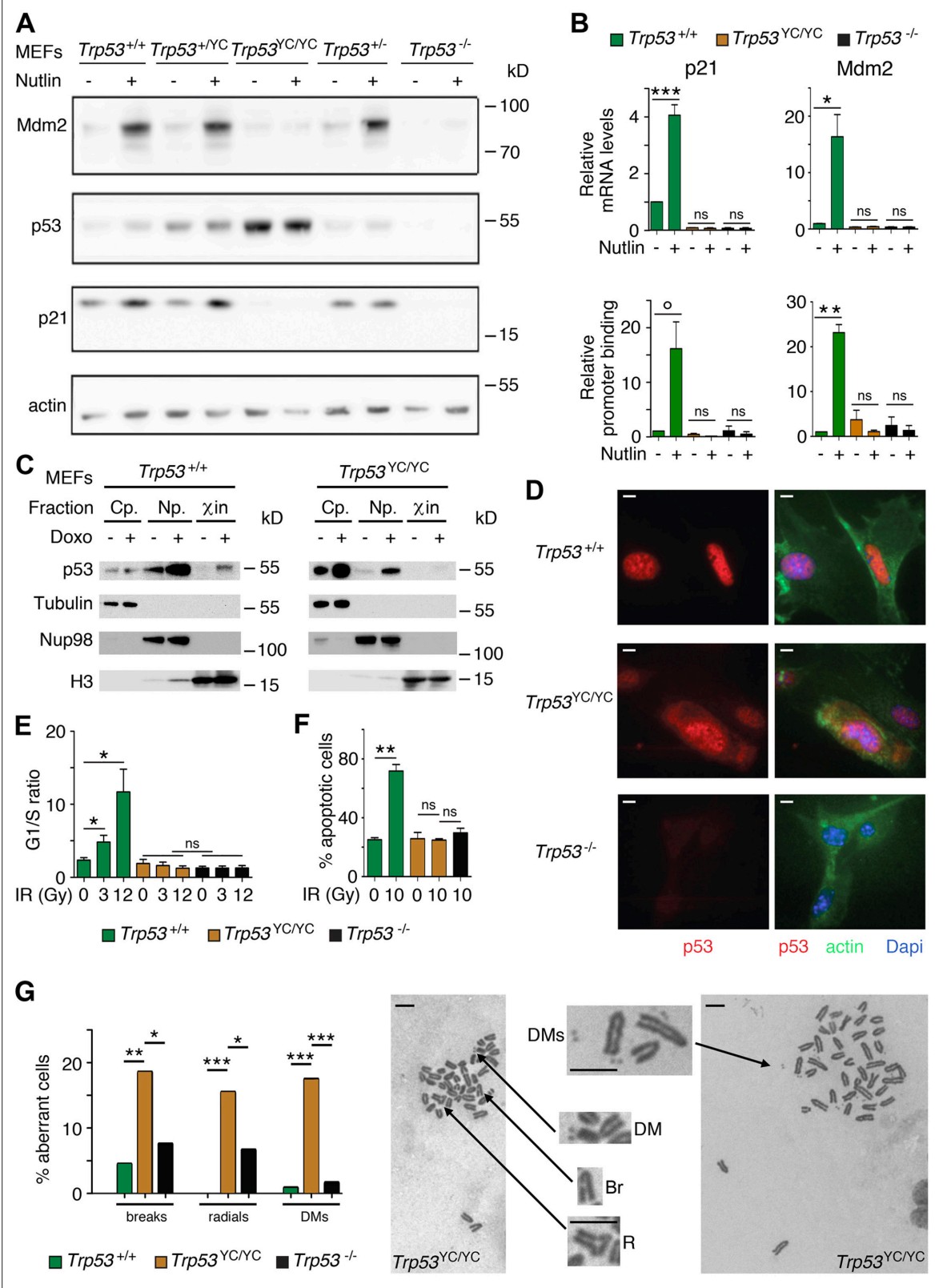

**Figure 2.** *Trp53*^YC/YC cells exhibit alterations in p53 abundance and subcellular localization, defective responses to DNA damage, and increased chromosomal instability. (**A**) Increased p53 protein levels in *Trp53*^YC/YC and *Trp53*^+/YC mouse embryonic fibroblasts (MEFs). MEFs of the indicated genotypes were treated or not with 10 μM Nutlin 3a for 24 hr, then protein extracts were immunoblotted with antibodies against Mdm2, p53, p21, and actin. (**B**) The transactivation of classical p53 target genes *Cdkn1a* and *Mdm2* is impaired in *Trp53*^YC/YC cells. Wildtype (WT), *Trp53*^YC/YC, and *Trp53*-

*Figure 2 continued*

$^{/-}$ MEFs were treated as in (**A**), then (top) mRNAs were quantified in five to six independent experiments using real-time PCR, with results normalized to control mRNAs and mean RNA amounts in unstressed WT cells assigned a value of 1; or (bottom) ChIP assays were performed at the *Cdkn1a* and *Mdm2* promoters in two to three independent experiments with an antibody against p53 or rabbit IgG as a negative control. Immunoprecipitates were quantified using real-time PCR, normalized to data over an irrelevant region, and the amount in unstressed WT cells was assigned a value of 1. Error bars: SEM. (**C**) Assessment of p53$^{WT}$ and p53$^{Y217C}$ subcellular localization by cellular fractionation. WT and *Trp53*$^{YC/YC}$ MEFs were treated or not with 1 µM doxorubicin (Doxo) for 24 hr, submitted to cellular fractionation, then protein extracts were immunoblotted with antibodies against p53 or the fraction controls Tubulin for cytoplasm (Cp.), Nup98 for nucleoplasm (Np.), and histone H3 for chromatin ($\chi$ in). (**D**) Assessment of p53$^{WT}$ and p53$^{Y217C}$ subcellular localization by immunofluorescence. WT, *Trp53*$^{YC/YC}$ and *Trp53*$^{-/-}$ MEFs were treated with 10 µM Nutlin 3a for 24 hr, then stained with antibodies against p53 (red) or actin (green) and DNA was counterstained with DAPI (blue). (**E**) Absence of a cell cycle arrest response in *Trp53*$^{YC/YC}$ MEFs. Asynchronous cell populations of *Trp53*$^{+/+}$, *Trp53*$^{YC/YC}$, and *Trp53*$^{-/-}$ MEFs were analyzed 24 hr after 0, 3, or 12 Gy γ-irradiation. Means + SEM from three independent experiments. (**F**) Absence of a p53-dependent apoptotic response in *Trp53*$^{YC/YC}$ thymocytes. Age-matched mice of the indicated genotypes were left untreated or submitted to 10 Gy whole-body γ-irradiation then sacrificed after 4 hr and their thymocytes were stained with Annexin V-FITC and analyzed by FACS. Means + SEM from two independent experiments. (**G**) Increased chromosomal instability in *Trp53*$^{YC/YC}$ fibroblasts. Metaphase spreads were prepared from WT, *Trp53*$^{YC/YC}$, and *Trp53*$^{-/-}$ MEFs at passage 4, then aberrant metaphases (with chromosome breaks, radial chromosomes, or double-minute chromosome [DMs]) were scored. Left: distribution of aberrant metaphases. Data from 110 WT, 97 *Trp53*$^{YC/YC}$, or 119 *Trp53*$^{-/-}$ complete diploid metaphases, independently observed by two experimenters. Right: examples of two aberrant *Trp53*$^{YC/YC}$ metaphases: one with a DM, a chromosome break (Br) and a radial chromosome (R), the other with multiple DMs. Enlargements of regions of interest are presented between the two metaphases. Scale bars (**D, G**): 5 µm. \*\*\*p<0.001, \*\*p<0.01, \*p<0.05, °p=0.09, ns: non-significant by Student's t (**B, E, F**) or Fisher's (**G**) tests.

The online version of this article includes the following source data and figure supplement(s) for figure 2:

**Source data 1.** Labeled files for gels and blots in *Figure 2A and C*.

**Source data 2.** Raw and unedited gels and blots for *Figure 2A and C*.

**Figure supplement 1.** Cell cycle arrest responses of *Trp53*$^{+/+}$, *Trp53*$^{YC/YC}$, and *Trp53*$^{-/-}$ mouse embryonic fibroblasts (MEFs).

**Figure supplement 2.** Apoptotic responses of *Trp53*$^{+/+}$, *Trp53*$^{-/-}$, and *Trp53*$^{YC/YC}$ thymocytes.

**Figure supplement 3.** Comparison of stress responses in wildtype (WT), *Trp53*$^{+/YC}$, and *Trp53*$^{+/-}$ cells.

On the opposite, we obtained clear evidence of a GOF for the p53$^{Y217C}$ mutant. Two p53 mutants (p53$^{G245D}$ and p53$^{R273H}$) were recently proposed to promote tumorigenesis by predisposing cells to chromosomal instability (*Zhao et al., 2024*), which led us to compare the frequency of chromosome rearrangements in WT, *Trp53*$^{YC/YC}$, and *Trp53*$^{-/-}$ primary MEFs. We searched for chromosome breaks, radial chromosomes, or double-minute chromosomes in preparations of MEFs treated with the anti-mitotic nocodazole for metaphase enrichment. The three categories of chromosome rearrangements were more frequently observed in *Trp53*$^{YC/YC}$ MEFs than in *Trp53*$^{-/-}$ or WT cells (*Figure 2G*), indicating a GOF that promotes chromosomal instability.

## Impact of p53$^{Y217C}$ on mouse development

We next determined the impact of the p53$^{Y217C}$ mutation in vivo, by comparing mouse cohorts generated from intercrosses of heterozygous *Trp53*$^{+/-}$ or *Trp53*$^{+/YC}$ mice resulting from ≥5 backcrosses to the C57BL/6J background. Intercrosses of *Trp53*$^{+/-}$ mice are known to yield a reduced proportion of *Trp53*$^{-/-}$ mice, that is mainly due to defects in neural tube closure causing exencephaly in a fraction of *Trp53*$^{-/-}$ female embryos (*Armstrong et al., 1995*; *Sah et al., 1995*), and, to a lesser extent, to lung dysfunction affecting a fraction of *Trp53*$^{-/-}$ neonates (*Tateossian et al., 2015*). At weaning (on the 21st day postpartum or P21), we observed one *Trp53*$^{-/-}$ female mouse for 3.5 *Trp53*$^{-/-}$ males from *Trp53*$^{+/-}$ intercrosses, an underrepresentation of females consistent with frequencies reported in earlier studies (*Armstrong et al., 1995*; *Sah et al., 1995*). Strikingly, the underrepresentation of weaned females was even more acute for *Trp53*$^{YC/YC}$ mice, with only one *Trp53*$^{YC/YC}$ female for 19 *Trp53*$^{YC/YC}$ males from *Trp53*$^{+/YC}$ intercrosses (*Figure 3A*). We next analyzed *Trp53*$^{YC/YC}$ embryos generated from heterozygous intercrosses or from heterozygous-homozygous (*Trp53*$^{+/YC}$ × *Trp53*$^{YC/YC}$) crosses, at embryonic days E12.5-E16.5. An underrepresentation of *Trp53*$^{YC/YC}$ female embryos was not observed, but 11/26 (42%) female embryos exhibited developmental abnormalities, including 10 with exencephaly, whereas all the male embryos appeared normal (*Figure 3B*). Importantly, the frequency of *Trp53*$^{YC/YC}$ female embryos with exencephaly (38.5%) was much higher than the reported frequency (0–8%) of *Trp53*$^{-/-}$ female embryos with exencephaly of C57BL/6J x 129/Sv genetic background (*Donehower et al., 1992*; *Sah et al., 1995*), suggesting a stronger effect of the p53$^{Y217C}$ mutant on female embryonic development.

**A**

**at weaning**

| *Trp53*$^{+/-}$ x *Trp53*$^{+/-}$ | Nbr. females obs (exp) | Nbr. males obs (exp) | Sex ratio (f/m) | Total |
|---|---|---|---|---|
| *Trp53*$^{+/+}$ | 24 (25) | 20 (25) | 1.2 | 196 |
| *Trp53*$^{+/-}$ | 59 (49) | 57 (49) | 1.04 | |
| *Trp53*$^{-/-}$ | 8 (25) | 28 (25) | 0.29 | |

| *Trp53*$^{+/YC}$ x *Trp53*$^{+/YC}$ | Nbr. females obs (exp) | Nbr. males obs (exp) | Sex ratio (f/m) | Total |
|---|---|---|---|---|
| *Trp53*$^{+/+}$ | 80 (85) | 88 (85) | 0.91 | 677 |
| *Trp53*$^{+/YC}$ | 209 (169) | 221 (169) | 0.95 | |
| *Trp53*$^{YC/YC}$ | 4 (85) | 75 (85) | 0.05 | |

**B**

**E12.5-E16.5 embryos**

| *Trp53*$^{+/YC}$ x *Trp53*$^{+/YC}$ or *Trp53*$^{+/YC}$ x *Trp53*$^{YC/YC}$ | Nbr. females | f exenc. (o.a.) | Nbr. males | m exenc. (o.a.) | Sex ratio (f/m) | Total |
|---|---|---|---|---|---|---|
| *Trp53*$^{+/+}$ | 12 | 0 (0) | 14 | 0 (0) | 0.86 | 169 |
| *Trp53*$^{+/YC}$ | 47 | 0 (0) | 48 | 0 (0) | 0.98 | |
| *Trp53*$^{YC/YC}$ | 26 | 10 (1) | 22 | 0 (0) | 1.18 | |

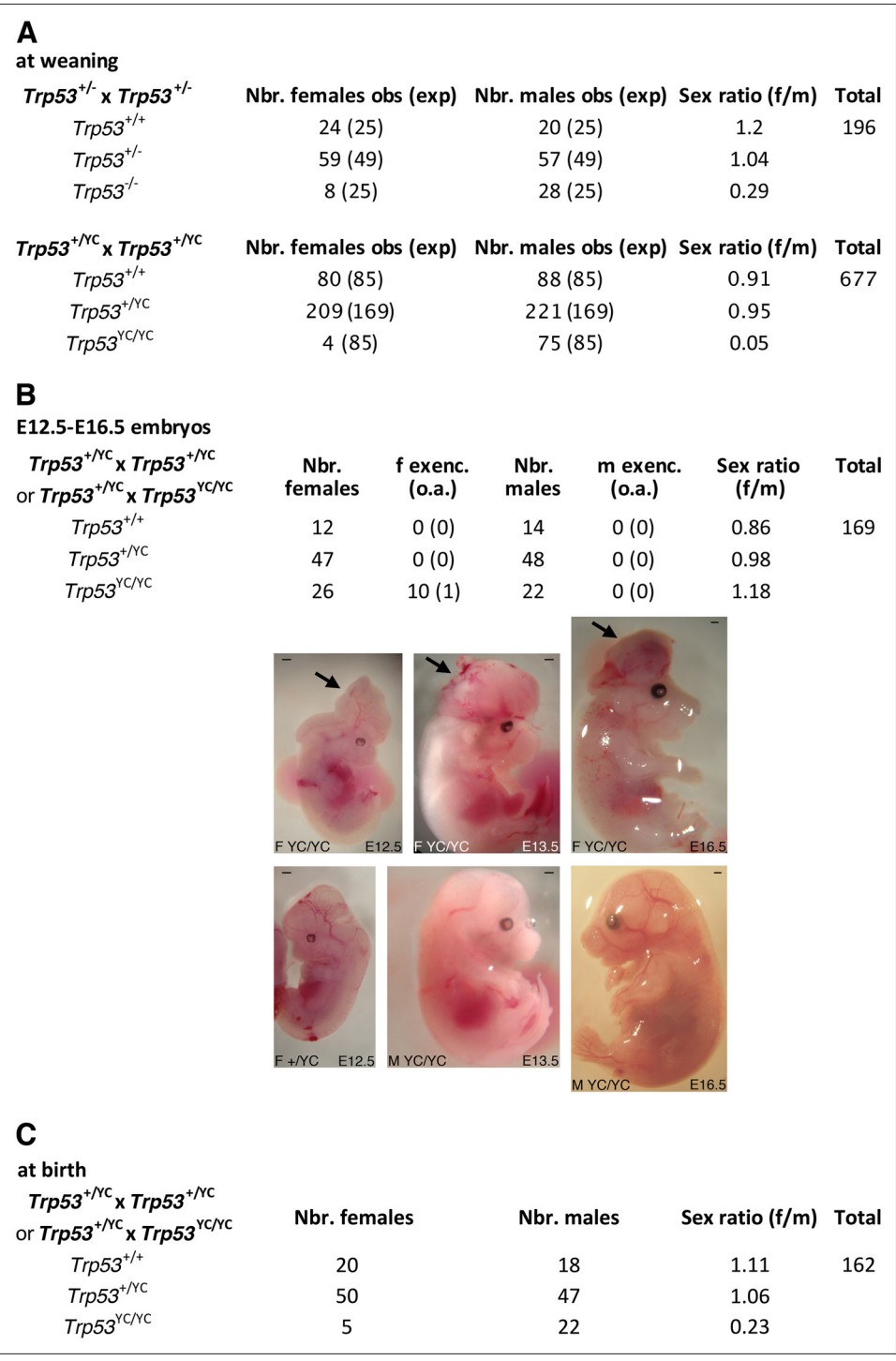

**C**

**at birth**

| *Trp53*$^{+/YC}$ x *Trp53*$^{+/YC}$ or *Trp53*$^{+/YC}$ x *Trp53*$^{YC/YC}$ | Nbr. females | Nbr. males | Sex ratio (f/m) | Total |
|---|---|---|---|---|
| *Trp53*$^{+/+}$ | 20 | 18 | 1.11 | 162 |
| *Trp53*$^{+/YC}$ | 50 | 47 | 1.06 | |
| *Trp53*$^{YC/YC}$ | 5 | 22 | 0.23 | |

**Figure 3.** *Trp53*$^{YC/YC}$ mice exhibit female-specific perinatal lethality. (**A**) Distribution of weaned mice obtained from *Trp53*$^{+/-}$ or *Trp53*$^{+/YC}$ intercrosses. Obs: observed numbers of mice at weaning (P21); exp: expected numbers assuming a Mendelian distribution without sex distortion; f/m: observed female/male ratios. Consistent with previous reports, the observed distribution of weaned mice from *Trp53*$^{+/-}$ intercrosses did not conform to values expected for a Mendelian distribution without sex distortion (U=5; $\chi^2$=16.31>15.09), indicating a significant deficit in female *Trp53*$^{-/-}$ mice (top). The distribution of weaned mice from *Trp53*$^{+/YC}$ intercrosses diverged even more from values for a Mendelian distribution without sex distortion (U=5; $\chi^2$=104.23>15.09), due to a striking deficit in female *Trp53*$^{YC/YC}$ mice (bottom). Differences between the frequencies of *Trp53*$^{YC/YC}$ (4/677) and *Trp53*$^{-/-}$ (8/196) females in the progeny, or between the female to male ratios for *Trp53*$^{YC/YC}$ (4/75) and *Trp53*$^{-/-}$ (8/28) animals, are statistically significant (p=0.0012 and p=0.0087 in Fisher's tests, respectively). (**B**) Exencephaly is frequently

*Figure 3 continued on next page*

*Figure 3 continued*

observed in p53$^{YC/YC}$ female embryos. Top: the distribution of E12.5–16.5 embryos from heterozygous (*Trp53$^{+/YC}$*) intercrosses or heterozygous-homozygous (*Trp53$^{+/YC}$ × Trp53$^{YC/YC}$*) crosses is shown. f or m exenc.: number of female or male embryos with exencephaly; o.a.: embryos with other abnormalities. Below, examples of female *Trp53$^{YC/YC}$* embryos at E12.5, E13.5, and E16.5 exhibiting exencephaly (arrows) are each shown (center) together with a normal embryo from the same litter (bottom). Scale bars : 1 mm. (**C**) Distribution of mice at birth from the indicated crosses. Of note, out of five *Trp53$^{YC/YC}$* females observed at birth, only one remained alive at weaning age. Thus, the female/male ratio for weaned *Trp53$^{YC/YC}$* animals from these crosses was 1/22, a ratio similar to the one observed in A (4/75).

The online version of this article includes the following figure supplement(s) for figure 3:

**Figure supplement 1.** Evidence of aberrant chromosome X inactivation in *Trp53$^{YC/YC}$* and *Trp53$^{-/-}$* female embryos.

In *Trp53$^{-/-}$* female mice, exencephaly was previously correlated with stochastic aberrant X chromosome inactivation, with a decreased expression of *Xist* and an increase in bi-allelic expression of X-linked genes including *Maob*, *Pls3*, and *Usp9x* (**Delbridge et al., 2019**). We prepared neurospheres from *Trp53$^{+/+}$*, *Trp53$^{-/-}$*, and *Trp53$^{YC/YC}$* female embryos and quantified mRNAs for these genes in neurospheres. Compared to neurospheres from WT animals, *Trp53$^{YC/YC}$* and *Trp53$^{-/-}$* neurospheres exhibited an apparent decrease in *Xist* expression and significantly higher levels of *Maob*, *Pls3*, and *Usp9x* (**Figure 3—figure supplement 1**). Thus, as for female *Trp53$^{-/-}$* mice, aberrant chromosome X inactivation may contribute to the underrepresentation of *Trp53$^{YC/YC}$* female mice. Of note however, defects in chromosome X inactivation did not appear more pronounced in *Trp53$^{YC/YC}$* neurospheres than in *Trp53$^{-/-}$* neurospheres.

**Table 1.** Viability of Mdm2 or Mdm4 loss in a *Trp53$^{YC/YC}$* background.

Mouse distributions from the indicated crosses were determined at weaning. As for *Trp53$^{+/YC}$* intercrosses, these crosses yielded a deficit in weaned *Trp53$^{YC/YC}$* females compared to *Trp53$^{YC/YC}$* males (4/75 from *Trp53$^{+/YC}$* intercrosses [**Figure 3A**]; 0/9 from crosses between *Mdm4$^{+/-}$ Trp53$^{+/YC}$* and *Mdm4$^{+/-}$ Trp53$^{YC/YC}$* mice; and 0/5 from mating *Mdm2$^{+/-}$ Trp53$^{+/YC}$* mice with *Mdm2$^{+/-}$ Trp53$^{YC/YC}$* or *Mdm2$^{-/-}$ Trp53$^{YC/YC}$* animals). *Trp53$^{YC/YC}$* females over *Mdm4* or *Mdm2* deficient or haploinsufficient backgrounds were also less frequent than their male counterparts, but the female/male ratio for *Trp53$^{YC/YC}$* mice over a *Mdm4$^{+/-}$* background (4/15) was significantly increased (p=0.04 when 4/75 and 4/15 ratios are compared in a Fisher's test). Whether or not the female/male ratios for *Trp53$^{YC/YC}$* mice were significantly increased over *Mdm4$^{-/-}$* (1/6), *Mdm2$^{-/-}$* (1/11), or *Mdm2$^{+/-}$* (1/11) backgrounds remained uncertain due to limited animal numbers.

| *Mdm4$^{+/-}$Trp53$^{YC/YC}$* × *Mdm4$^{+/-}$ Trp53$^{+/YC}$* | Nr. females | Nr. males | Sex ratio (f/m) | Total |
|---|---|---|---|---|
| *Mdm4$^{+/+}$Trp53$^{+/YC}$* | 8 | 11 | 0.73 | 88 |
| *Mdm4$^{+/+}$Trp53$^{YC/YC}$* | 0 | 9 | 0.00 | |
| *Mdm4$^{+/-}$Trp53$^{+/YC}$* | 15 | 19 | 0.79 | |
| *Mdm4$^{+/-}$Trp53$^{YC/YC}$* | 4 | 15 | 0.27 | |
| *Mdm4$^{-/-}$Trp53$^{+/YC}$* | 0 | 0 | NA | |
| *Mdm4$^{-/-}$Trp53$^{YC/YC}$* | 1 | 6 | 0.17 | |
| *Mdm2$^{+/-}$Trp53$^{YC/YC}$* × *Mdm2$^{+/-}$Trp53$^{+/YC}$* or *Mdm2$^{-/-}$ Trp53$^{YC/YC}$* × *Mdm2$^{+/-}$Trp53$^{+/YC}$* | Nr. females | Nr. males | Sex ratio (f/m) | Total |
| *Mdm2$^{+/+}$Trp53$^{+/YC}$* | 3 | 0 | NA | 63 |
| *Mdm2$^{+/+}$Trp53$^{YC/YC}$* | 0 | 5 | 0.00 | |
| *Mdm2$^{+/-}$Trp53$^{+/YC}$* | 16 | 15 | 1.07 | |
| *Mdm2$^{+/-}$Trp53$^{YC/YC}$* | 1 | 11 | 0.09 | |
| *Mdm2$^{-/-}$Trp53$^{+/YC}$* | 0 | 0 | NA | |
| *Mdm2$^{-/-}$Trp53$^{YC/YC}$* | 1 | 11 | 0.09 | |

To further analyze the impact of the p53[Y217C] mutation during development, we also determined its potential effect in embryos lacking Mdm2 or Mdm4, two major p53 regulators. The embryonic lethality resulting from Mdm2 or Mdm4 loss is rescued by a concomitant p53 deficiency (*Bardot et al., 2015*; *Finch et al., 2002*; *Jones et al., 1995*; *Migliorini et al., 2002*; *Montes de Oca Luna et al., 1995*; *Parant et al., 2001*). This provides a powerful assay for analyzing the functionality of p53 mutant alleles (*Bardot et al., 2015*; *Iwakuma et al., 2004*; *Marine et al., 2006*; *Toledo et al., 2006*). We identified *Mdm2*[-/-] *Trp53*[YC/YC] and *Mdm4*[-/-] *Trp53*[YC/YC] viable mice of both sexes, consistent with a major loss of canonical WT p53 activities in the p53[Y217C] mutant (*Table 1*). Of note, in these experiments we mated *Mdm4*[+/-] *Trp53*[+/YC] mice with *Mdm4*[+/-] *Trp53*[YC/YC] mice, or *Mdm2*[+/-] *Trp53*[+/YC] mice with either *Mdm2*[-/-] *Trp53*[YC/YC] or *Mdm2*[+/-] *Trp53*[YC/YC] animals, to analyze the progeny at weaning. No *Trp53*[YC/YC] female mouse was recovered from these crosses, whereas 14 *Trp53*[YC/YC] males were obtained, again demonstrating a striking deficit in *Trp53*[YC/YC] females at weaning. Among animals lacking one or two copies of either *Mdm2* or *Mdm4*, we also observed a deficit in *Trp53*[YC/YC] weaned females. However, the female to male ratio for *Trp53*[YC/YC] animals appeared markedly increased in genetic backgrounds with *Mdm4* haploinsufficiency (4/15) or loss (1/6), but only marginally increased (if at all) in genetic backgrounds with *Mdm2* haploinsufficiency (1/11) or loss (1/11). This suggested that altering the levels of p53 inhibitors, particularly Mdm4, might improve the survival of *Trp53*[YC/YC] females (*Table 1*).

The high frequency (42%) of abnormal *Trp53*[YC/YC] female embryos at E12.5-E16.5 could only partially account for the acute deficit in *Trp53*[YC/YC] females observed at weaning, suggesting either embryonic abnormalities undetectable macroscopically, or that a fraction of *Trp53*[YC/YC] females died later in development or postpartum. We performed additional crosses to analyze female to male ratios at postpartum day 0 or 1 (P0-P1) and observed 20 females for 18 males for *Trp53*[+/+] animals, but only 5 females for 22 males for *Trp53*[YC/YC] animals (*Figure 3C*). Furthermore, only one of these *Trp53*[YC/YC] females reached weaning age, and the four other neonates died before P2. Altogether, these data led to conclude that the p53[Y217C] mutation caused female-specific developmental abnormalities at a higher penetrance than a null allele, leading to a perinatal (late embryonic or early postpartum) lethality for most homozygous mutant females.

Interestingly, we observed that *Trp53*[YC/YC] males were fertile and included them in some of our crosses to generate homozygous mutants at higher frequencies (*Figure 3B and C*). By contrast, we were able to mate two *Trp53*[YC/YC] adult females with a *Trp53*[+/YC] male, and both got pregnant but had to be euthanized due to complications while giving birth. In both cases, extended labor (>24 hr)

**Table 2.** Evidence of dystocia in pregnant *Trp53*[YC/YC] females.

Results of the mating of two (a and b) *Trp53*[YC/YC] females (F) with a *Trp53*[+/YC] male (M) are shown. Both females were rapidly pregnant after encountering a male, but had to be sacrificed due to extended labor and pain during their first (female a) or third (female b) delivery.

| F *Trp53*[YC/YC] a<br>mated with M *Trp53*[+/YC] | |
| --- | --- |
| | Litter 1 |
| | 1 pup born 23 days after mating was initiated, found dead (partially eaten) |
| | 24 hr later, mother is still in labor |
| | 48 hr later, another pup found dead in cage, the mother appeared in pain and was sacrificed |
| **F *Trp53*[YC/YC] b<br>mated with M *Trp53*[+/YC]** | |
| | Litter 1 |
| | 2 pups born 24 days after mating was initiated: 1 found dead, 1 alive (F *Trp53*[+/YC]) |
| | Litter 2 |
| | 4 pups, all alive: 1 F *Trp53*[+/YC], 3 M *Trp53*[+/YC] |
| | Litter 3 |
| | After extended labor (24 hr) pup fragments were found in cage, the mother in pain was sacrificed with pups still inside the womb |

was the main sign of dystocia (*Table 2*). These observations suggest that, even for the rare *Trp53*<sup>YC/YC</sup> females able to reach adulthood, the p53<sup>Y217C</sup> mutation caused pathological traits not observed in *Trp53*<sup>-/-</sup> females, because the loss of p53 was not previously reported to cause dystocia (*Embree-Ku and Boekelheide, 2002*; *Guimond et al., 1996*; *Hu et al., 2007*). Together, our data indicated that the p53<sup>Y217C</sup> mutation not only abolished canonical p53 activities, but also conferred additional effects compromising female perinatal viability or parturition.

## Impact of p53<sup>Y217C</sup> on tumorigenesis

We next analyzed the impact of the p53<sup>Y217C</sup> mutation on the onset and spectrum of spontaneous tumors in mice. We first compared *Trp53*<sup>+/YC</sup>, *Trp53*<sup>+/-</sup>, and *Trp53*<sup>+/+</sup> mouse cohorts for 2 years. *Trp53*<sup>+/YC</sup> and *Trp53*<sup>+/-</sup> cohorts exhibited similar spontaneous tumor onset and spectrum, with most mice developing sarcomas during their second year of life (*Figure 4—figure supplement 1*). Thus, at least for tumors arising spontaneously, the p53<sup>Y217C</sup> mutant protein did not appear to exert a DNE over the WT p53 protein.

We next compared *Trp53*<sup>YC/YC</sup> and *Trp53*<sup>-/-</sup> mouse cohorts. Because of the difficulty to generate *Trp53*<sup>YC/YC</sup> females, we restricted our comparison to males, to avoid potential biases that might result from different sex ratios. All *Trp53*<sup>-/-</sup> mice are known to die in less than 1 year, from thymic lymphomas in most cases, or more rarely from sarcomas (*Donehower et al., 1992*; *Jacks et al., 1994*). *Trp53*<sup>YC/YC</sup> males died faster than their *Trp53*<sup>-/-</sup> counterparts: all the *Trp53*<sup>YC/YC</sup> males were dead by the age of 7 months, whereas more than 20% of the *Trp53*<sup>-/-</sup> males were still alive at that age (*Figure 4A*). Furthermore, most *Trp53*<sup>-/-</sup> and *Trp53*<sup>YC/YC</sup> mice died from thymic lymphomas, but histological analysis of tumor organs revealed that the lymphomas in *Trp53*<sup>YC/YC</sup> males were more aggressive and invasive, with sites of metastases notably including the lungs, spleen, liver, or kidneys (*Figure 4B and C*, *Figure 4—figure supplement 2*). Thus, the p53<sup>Y217C</sup> mutation not only abolished canonical tumor suppressive activities but also apparently conferred oncogenic properties to the encoded protein. Consistent with this, when we performed an RNA-seq analysis of thymi from 8-week-old *Trp53*<sup>+/+</sup>, *Trp53*<sup>-/-</sup>, and *Trp53*<sup>YC/YC</sup> males, 81.5% of the 717 differentially expressed genes indicated an LOF in the p53<sup>Y217C</sup> mutant, but genes suggesting a GOF or an SOF were also observed (*Figure 4—figure supplement 3A* and *Supplementary file 1*). Of note, among the differentially expressed genes corresponding to an LOF in the p53<sup>Y217C</sup> mutant were *Bbc3* (alias *Puma*), *Cdkn1a* (alias *p21*), and *Zmat3* (*Figure 4—figure supplement 3B*), three p53 target genes known to play major roles in p53-mediated tumor suppression (*Brennan et al., 2024*).

We next performed a comparative analysis focusing on *Trp53*<sup>YC/YC</sup> and *Trp53*<sup>-/-</sup> thymi, which revealed 192 differentially expressed genes (*Figure 5A and B* and *Supplementary file 2*). An analysis of these data with GOrilla, the Gene Ontology enRIchment anaLysis and visuaLizAtion tool (*Eden et al., 2009*), indicated that 141 of these genes were associated with at least one gene ontology (GO) term, and revealed a significant enrichment for genes associated with white blood cell chemotaxis/migration and inflammation (*Figure 5C*, *Figure 5—figure supplement 1*). Among these genes were notably *Ccl17*, *Ccl9*, *Ccr3*, *Cxcl10*, *S100a8*, and *S100a9*, six genes associated each with 10–15 GO terms related to white blood cell behavior and inflammation (*Supplementary file 3*). We next used RT-qPCR to quantify the expression of these genes in thymi from 8-week-old *Trp53*<sup>+/+</sup>, *Trp53*<sup>YC/YC</sup>, and *Trp53*<sup>-/-</sup> male mice. Their expression was significantly increased in *Trp53*<sup>YC/YC</sup> thymic cells compared to *Trp53*<sup>-/-</sup>, or to both *Trp53*<sup>+/+</sup> and *Trp53*<sup>-/-</sup> cells, consistent with GOF/SOF effects in the p53<sup>Y217C</sup> mutant (*Figure 5D*). We also compared the transcriptomes of *Trp53*<sup>YC/YC</sup> and *Trp53*<sup>-/-</sup> thymocytes by gene set enrichment analysis (GSEA) (*Subramanian et al., 2005*). We found 13 gene sets significantly enriched in *Trp53*<sup>YC/YC</sup> cells with normalized enrichment scores (NES)>2, among which three ('antimicrobial peptides', 'chemokine receptors bind chemokines', and 'defensins') were related to immunity (*Figure 5E*, *Figure 5—figure supplement 2A*), consistent with an inflammatory response associated with the p53<sup>Y217C</sup> mutant. Other enriched gene sets notably included five sets ('electron transport chain', 'respiratory electron transport ATP synthesis by chemiosmotic coupling and heat production by uncoupling proteins', 'mitochondrial translation', 'respiratory electron transport', and 'oxidative phosphorylation') related to mitochondria function, and three sets ('deposition of new CENPA-containing nucleosomes at the centromere', 'arginine methyltransferases methylate histone arginines', and 'PRC2 methylates histones and DNA') related to chromatin plasticity (*Figure 5E*, *Figure 5—figure supplement 2B and C*). These gene sets, differentially expressed between *Trp53*<sup>YC/YC</sup> and *Trp53*<sup>-/-</sup> cells, might contribute to

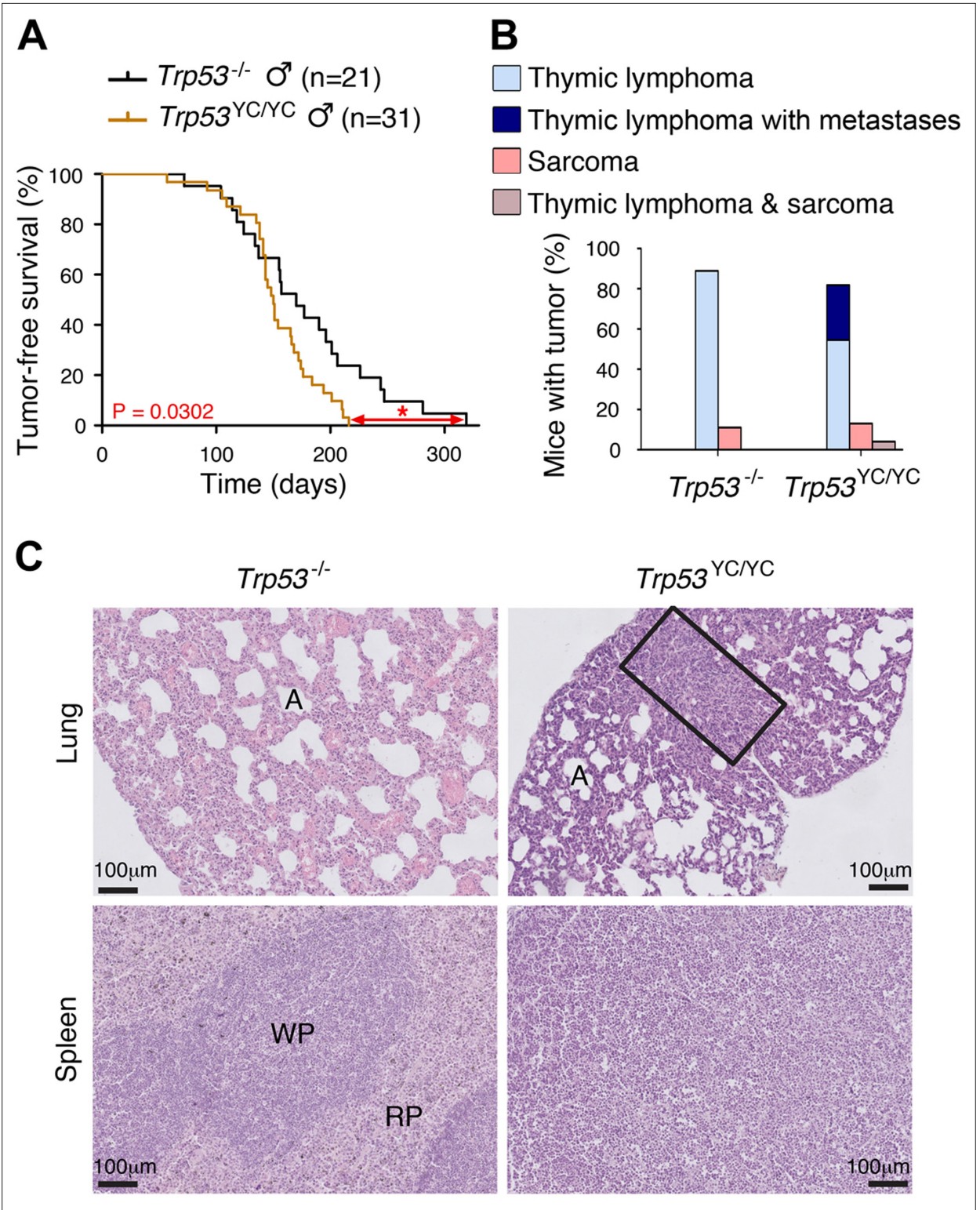

**Figure 4.** Oncogenic effects of the mutant protein in *Trp53*^YC/YC^ male mice. (**A–B**) In homozygous males, p53^Y217C^ leads to accelerated tumor-induced death (**A**), and aggressive metastatic tumors (**B**); n=cohort size. (**C**) Hematoxylin and eosin (H&E) staining of sections from the lung (top) and spleen (bottom) of *Trp53*^-/-^ and *Trp53*^YC/YC^ male mice, showing metastases in *Trp53*^YC/YC^ animals. Normal organ structures are shown, with 'A' indicating pulmonary alveoli, and 'WP' and 'RP' standing for splenic white and red pulp, respectively. In the lung section of the *Trp53*^YC/YC^ mouse, the rectangle indicates a lymphoma area. In the spleen section of the *Trp53*^YC/YC^ mouse, the typical splenic structures are absent due to massive tissue homogenization of the spleen by lymphoma cells.

*Figure 4 continued on next page*

*Figure 4 continued*

The online version of this article includes the following figure supplement(s) for figure 4:

**Figure supplement 1.** *Trp53*[+/YC] and *Trp53*[+/-] mice exhibit similar tumor onset and spectra.

**Figure supplement 2.** Example of a *Trp53*[YC/YC] mouse with a thymic lymphoma and lymphomatous infiltrates in the liver and kidney.

**Figure supplement 3.** RNA-seq analysis from the thymi of 8-week-old *Trp53*[+/+], *Trp53*[YC/YC], and *Trp53*[-/-] male mice.

accelerated tumorigenesis in *Trp53*[YC/YC] mice, given the reported impact of inflammation, metabolism changes, and epigenetic plasticity on cancer evolution (*Feinberg et al., 2006*; *Greten and Grivennikov, 2019*; *Kroemer and Pouyssegur, 2008*). Furthermore, the hotspot mutants p53[G245D] or p53[R273H] were recently proposed to promote tumor progression by inducing non-canonical nuclear factor kappa light chain enhancer of activated B cell (NC-NF-κB) signaling (*Zhao et al., 2024*). In *Trp53*[YC/YC] cells, one gene set related to NC-NF-κB ('Dectin 1-mediated non-canonical NF-κB signaling') was significantly enriched and two other NF-κB-related gene sets were also potentially enriched (*Figure 5F*, *Figure 5—figure supplement 3*).

## Reducing inflammation may rescue a fraction of female *Trp53*[YC/YC] embryos

Our data indicated that inflammation correlated with accelerated tumorigenesis in *Trp53*[YC/YC] male mice. Interestingly, inflammation was previously proposed to promote neural tube defects or embryonic death in a few mouse models (*Lian et al., 2004*; *McNairn et al., 2019*; *Zhao et al., 2013*). Furthermore *CD44*, encoding a cell-surface glycoprotein that drives inflammation and cancer progression (*Solier et al., 2023*), was recently identified as a key gene for the diagnosis and early detection of open neural tube defects (*Karthik et al., 2022*). Together, these data led us to hypothesize that, at least in a fraction of *Trp53*[YC/YC] females, neural tube defects might result from inflammation. To test this hypothesis, we mated *Trp53*[+/YC] female mice with *Trp53*[YC/YC] males, then administered supformin to pregnant females by oral gavage. Supformin (LCC-12), a potent inhibitor of the CD44-copper signaling pathway, was shown to reduce inflammation in vivo (*Solier et al., 2023*). Without any treatment, our previous crosses (*Figure 3* and *Table 1*) collectively yielded a female to male ratio of 0.045 for *Trp53*[YC/YC] weaned mice (f/m=5/111 [4+1+ 0+0]/[75+22+ 9+5]), significantly different from the ratio of 0.29 (f/m=8/28) for *Trp53*[-/-] weaned animals. By contrast, in the progeny of supformin-treated pregnant mice we observed a fivefold increase in the female to male ratio for *Trp53*[YC/YC] weaned animals, to reach a value of 0.23 (f/m=3/13) indistinguishable from the ratio in *Trp53*[-/-] weaned animals from untreated mice (*Figure 5G* and *Supplementary file 4*). This result suggests that reducing inflammation in developing *Trp53*[YC/YC] female embryos may increase their viability.

## Discussion

In this report, we generated a mouse model with a *Trp53*[Y217C] allele, the murine equivalent of the human hotspot mutant *TP53*[Y220C]. The analysis of this mouse model indicated that the p53[Y217C] mutation leads to the loss of many canonical WT p53 activities, notably the capacity to transactivate target genes important for tumor suppression (e.g. *Bbc3*, *Cdkn1a*, and *Zmat3*). The fact that p53[Y217C] can rescue the embryonic lethality caused by a deficiency in Mdm2 or Mdm4 is also consistent with a major loss of canonical WT p53 functions (LOF) in this mutant. These findings are consistent with the notion that hotspot p53 mutants cause a complete or near complete LOF (*Funk et al., 2025*).

In addition, our analyses provide evidence that the p53[Y217C] mutation exhibits oncogenic effects. The possibility that a mutant p53 might acquire oncogenic functions was first suggested over 30 years ago (*Dittmer et al., 1993*), but this notion became controversial in recent years. Indeed, studies of a few p53 mutants in human acute myeloid leukemia and a mouse model of B cell lymphoma indicated that their increased tumorigenicity might result from DNE rather than GOF, and a recent study further suggested that the putative GOF of many p53 mutants would not be required to sustain the expansion of tumors (*Aubrey et al., 2018*; *Boettcher et al., 2019*; *Wang et al., 2024*). Here, however, we observed accelerated tumorigenesis in *Trp53*[YC/YC] mice, but did not observe any evidence of DNE for spontaneous tumorigenesis in *Trp53*[+/YC] animals. Importantly, the fact that p53[Y217C] did not exhibit a DNE over the WT protein is consistent with the report that Li-Fraumeni patients carrying

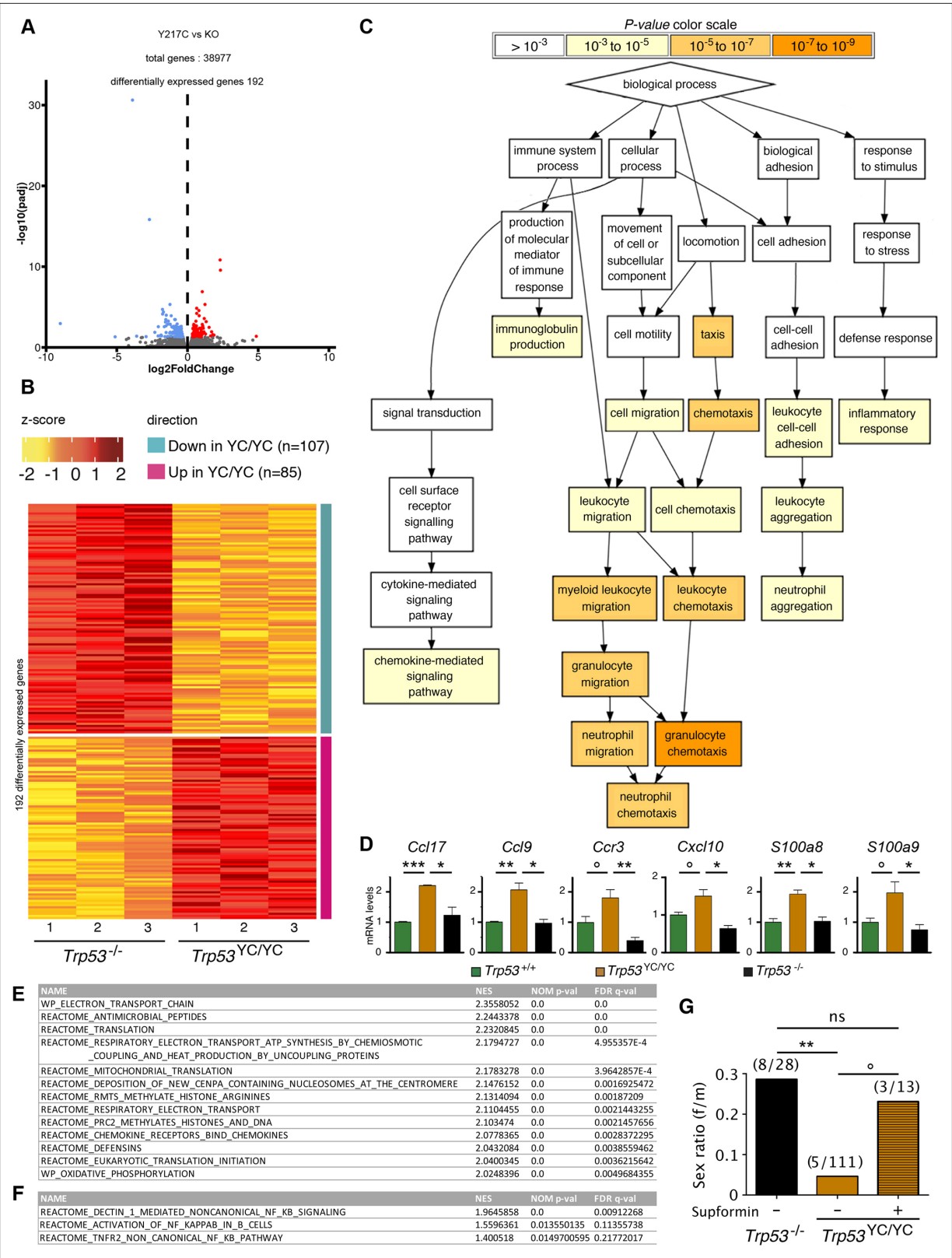

**Figure 5.** Evidence of inflammation in *Trp53*^YC/YC mice. (**A–B**) RNA-seq analysis of thymi from *Trp53*^YC/YC (n=3) and *Trp53*^-/- (n=3) 8-week-old male mice. Volcano plot (**A**), with differentially expressed genes (DEGs), downregulated (blue) or upregulated (red) in *Trp53*^YC/YC cells. Unsupervised clustering heat-map plot (**B**), with 192 DEGs ranked according to log₂ fold changes. (**C**) Gene ontology (GO) analysis of DEGs. Out of 192 DEGs, 141 are associated with at least one GO term, according to the Gene Ontology enRIchment anaLysis and visuaLizAtion tool (GOrilla). For each GO term, enrichment was

*Figure 5 continued on next page*

*Figure 5 continued*

calculated by comparing with the full list of 19,759 genes associated with a GO term, and results are presented here with a color scale according to p-value of enrichment. This analysis mainly revealed an enrichment for genes associated with white blood cell chemotaxis/migration and inflammation, as shown here. Complete results of the analysis are presented in *Figure 5—figure supplement 1*. (D) RT-qPCR analysis of the indicated genes in thymi from 8-week-old *Trp53*[+/+], *Trp53*[YC/YC], and *Trp53*[-/-] male mice. Means + SEM (n=3 per genotype). \*\*\*p<0.001, \*\*p<0.01, \*p<0.05, °p<0.075 by Student's t-test. (E) Gene set enrichment analysis (GSEA) of transcriptomes from *Trp53*[YC/YC] and *Trp53*[-/-] thymic cells. GSEA identified 13 gene sets enriched in *Trp53*[YC/YC] cells with normalized enrichment scores (NES)>2. Nominal p-values (NOM p-val) and false discovery rate q-values (FDR q-val) are indicated. (F) GSEA provides evidence of increased NC-NF-κB signaling in *Trp53*[YC/YC] male thymocytes. Results are presented as in (E). (G) Supformin (LCC-12), an anti-inflammatory molecule, increases the female to male ratio in *Trp53*[YC/YC] weaned pups. *Trp53*[+/YC] females were mated with *Trp53*[YC/YC] males, then oral gavage of pregnant females with 5 mg/kg supformin was performed on the 10th and 11th days post-coitum, and their progeny was genotyped at weaning. The female to male (f/m) ratio for *Trp53*[YC/YC] weaned pups that were exposed to supformin in utero (+) was compared to the f/m ratios for *Trp53*[-/-] or *Trp53*[YC/YC] weaned pups never exposed to supformin (-). Exposure to supformin led to a fivefold increase in the proportion of *Trp53*[YC/YC] weaned females. \*\*p<0.01, °p=0.056, ns: non-significant by Fisher's test.

The online version of this article includes the following figure supplement(s) for figure 5:

**Figure supplement 1.** Gene ontology (GO) analysis of differentially expressed genes (DEGs).

**Figure supplement 2.** Gene set enrichment analysis (GSEA) of transcriptomes from *Trp53*[Y217C/Y217C] (Y220C) and *Trp53*[-/-] (KO) thymic cells: examples of GSEA enrichment plots.

**Figure supplement 3.** Gene set enrichment analysis (GSEA) enrichment plot showing increased NC-NF-κB signaling in *Trp53*[YC/YC] cells.

an heterozygous *TP53*[Y220C] mutation display a similar age of cancer onset than Li-Fraumeni patients with an heterozygous non-sense *TP53* mutation (*Xu et al., 2014*). Furthermore, our evidence of an oncogenic GOF leading to aggressive metastatic tumors in *Trp53*[YC/YC] male mice is consistent with the oncogenic GOF attributed to p53[Y220C] in male patients with glioblastoma (*Rockwell et al., 2021*) or with experiments suggesting that p53[Y220C] expression in p53-null cells, or its overexpression in MCF-10A cells, may increase their migratory or invasive capacities (*Pal et al., 2023*; *Zhou et al., 2022*).

Our transcriptomic analyses suggested that the p53[Y217C] mutant might accelerate tumorigenesis by promoting inflammation, a hallmark of cancer (*Hanahan, 2022*) previously associated with a few other hotspot p53 mutants with oncogenic GOF (*Agupitan et al., 2020*; *Behring et al., 2019*; *Ham et al., 2019*; *Zhao et al., 2024*). Interestingly, the GSEA comparison of transcriptomes from *Trp53*[YC/YC] and *Trp53*[-/-] male thymic cells indicated differences related to inflammation and chemokine signaling, but also to mitochondria function and chromatin plasticity, and a link between inflammation, mitochondrial copper, and epigenetic plasticity was recently demonstrated (*Solier et al., 2023*). Furthermore, the p53[G245D] or p53[R273H] mutants were proposed to accelerate tumorigenesis by promoting a chromosomal instability that would activate NC-NF-κB signaling and promote tumor cell metastasis (*Zhao et al., 2024*). Here, we observed increased chromosomal instability in *Trp53*[YC/YC] fibroblasts, increased NC-NF-κB and inflammation-related signaling in *Trp53*[YC/YC] thymocytes, and increased tumor metastasis in *Trp53*[YC/YC] mice compared to their *Trp53*[-/-] counterparts. These data suggest that similar mechanisms might underlie the oncogenic properties of the p53[Y217C], p53[G245D], and p53[R273H] mutants.

A striking and unexpected effect of the p53[Y217C] mutation was its severe impact on the perinatal viability of female mice, which led to observe only four *Trp53*[YC/YC] females for 75 *Trp53*[YC/YC] males at weaning from heterozygous intercrosses, or five *Trp53*[YC/YC] females for 111 *Trp53*[YC/YC] males when all weaned animals from relevant crosses were considered. The perinatal lethality of *Trp53*[YC/YC] females correlated with a high frequency (38.5%) of exencephalic females at E12.5–16.5 embryonic ages. By comparison, the female to male ratio at weaning was 8/28 for *Trp53*[-/-] mice, and only 0–8% of exencephalic female *Trp53*[-/-] embryos were reported in similar genetic backgrounds (*Armstrong et al., 1995*; *Donehower et al., 1992*; *Sah et al., 1995*). These observations provide evidence of teratogenic effects for the p53[Y217C] mutant protein. Three other mouse p53 models were previously found to cause exencephaly at a higher frequency than a p53 null allele: *Trp53*[NLS1], with three mutations at residues 316–318 affecting a nuclear localization signal and leading to a predominantly cytoplasmic localization of p53[NLS1] in most cells (*Regeling et al., 2011*); *Trp53*[N236S], a mouse model of *TP53*[N239S], a recurrent but uncommon mutant in human cancers (*Zhao et al., 2019*); and *Bim*[+/-] *Trp53*[-/-] mice, combining p53 loss with an haploinsufficiency in the cytoplasmic proapoptotic regulator Bim/Bcl2l11 (*Delbridge et al., 2019*). The female-specific lethality of *Trp53*[N236S] mice was proposed to result from increased *Xist* expression, whereas a decrease in *Xist* expression was observed in *Bim*[+/-] *Trp53*[-/-] mice, suggesting distinct causes for neural tube defects (*Delbridge et al., 2019*; *Zhao et al., 2019*). We

observed decreased *Xist* expression and an increased expression of X-linked genes in neurospheres from *Trp53*[YC/YC] females compared to WT females, whereas these genes were expressed at similar levels in neurospheres from *Trp53*[YC/YC] and *Trp53*[-/-] females. These results are not consistent with the mechanism proposed for exencephaly in the *Trp53*[N236S] mouse model, but might rather reflect a p53 LOF in *Trp53*[YC/YC] embryos. In addition, the analyses of the *Trp53*[NLS1] and *Bim*[+/-] *Trp53*[-/-] mouse models make it tempting to speculate that, in female *Trp53*[YC/YC] embryos, the abundance of p53[Y217C] in the cytoplasm might perturb mitochondria function (*Blandino et al., 2020*) to further promote neural tube closure defects. Consistent with this possibility, p53 mutant proteins accumulating in the cytoplasm were reported to inhibit autophagy (*Morselli et al., 2008*), a key determinant for mitochondrial integrity (*Rambold and Lippincott-Schwartz, 2011*), and autophagy impairment may promote neural tube defects (*Fimia et al., 2007*; *Ye et al., 2020*).

Importantly, our data suggest that common mechanisms might contribute to the oncogenic effects accelerating tumorigenesis in *Trp53*[YC/YC] males and the teratogenic effects causing the perinatal lethality of many *Trp53*[YC/YC] females. Indeed, the RNA-seq analysis of male thymocytes indicated that inflammation likely promotes oncogenesis in *Trp53*[YC/YC] males, and the administration of the anti-inflammatory drug supformin to pregnant mice increased the ratio of *Trp53*[YC/YC] weaned females in their progeny. Thus, the low female/male ratio observed for *Trp53*[YC/YC] weaned animals would likely result not only from aberrant X chromosome inactivation in female embryos as in *Trp53*[-/-] animals (*Delbridge et al., 2019*), but also from inflammation. Presumably, a higher level of chromosomal instability in *Trp53*[YC/YC] cells might promote inflammation in *Trp53*[YC/YC] embryos (*Rodier et al., 2009*), and this might be more detrimental to females because *Trp53*[YC/YC] males would be protected by the anti-inflammatory effects of androgens (*Ainslie et al., 2024*; *Bianchi, 2019*; *McNairn et al., 2019*). Of note, we also observed that decreased Mdm4 levels increased the female/male ratio in *Trp53*[YC/YC] weaned mice. Because MDM4 overexpression was shown to promote genome instability and inflammation in cells independently of p53 (*Carrillo et al., 2015*; *Liu et al., 2024*), it is tempting to speculate that, as for supformin-treated mice, the improved survival of *Trp53*[YC/YC] female embryos in an Mdm4[+/-] background might result from decreased inflammation. Interestingly, in the offspring of human pregnancies at risk of early preterm delivery, an interleukin-6 polymorphism was associated with a female-specific susceptibility to adverse neurodevelopmental outcome (*Varner et al., 2020*), suggesting that inflammation may also impact the development of human female embryos. Finally, our mating attempts with *Trp53*[YC/YC] mice indicated that *Trp53*[YC/YC] pregnant females exhibited dystocia, and parturition is considered to be a finely regulated inflammatory process (*Zhang and Wei, 2021*). Of note, dystocia was not observed in *Trp53*[-/-] mice, but was previously reported in pregnant females with a combined deficiency of p53 and FasL (*Embree-Ku and Boekelheide, 2002*). The fact that *Trp53*[YC/YC] females shared phenotypic traits with both *Bim*[+/-] *Trp53*[-/-] mice (neural tube defects) and *FasL*[-/-] *Trp53*[-/-] mice (dystocia) may seem relevant, because Bim and FasL are both regulators of apoptosis and autophagy that also regulate immune responses (*Sionov et al., 2015*; *Taylor and Ng, 2018*). Altogether, these data suggest that inflammation may cause accelerated tumorigenesis in

**Table 3.** Effects of the *Trp53*[Y217C] mutation: a summary.

The comparison between *Trp53*[-/-] and *Trp53*[Y217C/Y217C] mice is presented. The phenotypes observed in *Trp53*[-/-] male (M) and female (F) mice result from p53 loss of function (LOF), whereas those observed in *Trp53*[Y217C/Y217C] mice result from p53 LOF as well as additional effects (gain of function [GOF] in red). The + signs denote the presence of a phenotype. Xi: X chromosome inactivation.

| *Trp53*[-/-] (LOF) | *Trp53*[Y217C/Y217C] (LOF+GOF) |
|---|---|
| Spontaneous tumors (mostly thymic lymphomas), death in ≤1 year. | Spontaneous tumors (mostly thymic lymphomas) with metastases, death in ≤7 months |
| ++ <br> (M & F) | +++ <br> (only M were evaluated due to scarcity of F) inflammation in thymic cells |
| Perinatal lethality <br> + <br> (F only: Xi aberrations) | Perinatal lethality increased <br> ++++ <br> (F only: Xi aberrations and inflammation) <br> F partial rescue by supformin |
| No dystocia <br><br> (pregnant F) | Dystocia <br> + <br> (pregnant F) |

*Trp53*<sup>YC/YC</sup> male mice and contribute to promote neural tube defects in *Trp53*<sup>YC/YC</sup> female embryos or dystocia in *Trp53*<sup>YC/YC</sup> pregnant females (*Table 3*).

In conclusion, we generated a mouse model expressing p53<sup>Y217C</sup>, the murine equivalent of human p53<sup>Y220C</sup>, and its analysis revealed that the GOF of a hotspot mutant p53 may not be limited to onco-genic effects. This is an important notion to consider given the ever-expanding functions ascribed to p53 (*Hu, 2009*; *Rakotopare and Toledo, 2023*; *Voskarides and Giannopoulou, 2023*; *Vousden and Lane, 2007*). In addition, our results unveiled the potential relevance of supformin in obstetrics, which deserves further investigation.

# Materials and methods

**Key resources table**

| Reagent type (species) or resource | Designation | Source or reference | Identifiers | Additional information |
|---|---|---|---|---|
| Gene (*Mus musculus*) | *Trp53* | GenBank | ENSMUSG00000059552 | |
| Strain, strain background (*M. musculus*, both sexes) | *Trp53*<sup>Y217C</sup> (*Trp53*<sup>YC</sup>), C57Bl/6J | This paper | | 'Targeting in ES cells and genotyping' |
| Strain, strain background (*M. musculus*, females) | *PGK-Cre*, C57Bl/6J | Jackson Labs | B6.C-Tg(Pgk1-cre)1Lni/CrsJ | |
| Strain, strain background (*M. musculus*, both sexes) | *Trp53*<sup>-</sup>, C57Bl/6J | Jackson Labs | B6.129S2-*Trp53*<sup>tm1Tyj</sup>/J | |
| Strain, strain background (*M. musculus*, both sexes) | C57Bl/6J | Charles River Labs | | |
| Cell line (*M. musculus*, both sexes) | WT, *Trp53*<sup>+/YC</sup>, *Trp53*<sup>YC/YC</sup>, *Trp53*<sup>+/-</sup>, *Trp53*<sup>-/-</sup> fibroblasts | This paper | Primary fibroblasts from E13.5 embryos | 'Cells and cell culture reagents' |
| Cell line (*M. musculus*, females) | WT, *Trp53*<sup>YC/YC</sup>, *Trp53*<sup>-/-</sup> neurospheres | This paper | Neurospheres from E14.5 embryos | 'Cells and cell culture reagents' |
| Antibody | p53 (Rabbit polyclonal) | Novocastra | Leica NCL-p53-CM5p | 1/2000 |
| Antibody | p53 (Rabbit polyclonal) | Santa Cruz | FL-393 Sc 6243 | 50 µg |
| Antibody | p53 (Goat polyclonal) | R&D Systems | AF-1355 | 1/600 |
| Antibody | Mdm2 (Mouse monoclonal) | Abcam | 4B-2 | 1/500 |
| Antibody | p21 (Mouse monoclonal) | Santa Cruz | F-5 sc6246 | 1/200 |
| Antibody | Actin (Mouse monoclonal) | Santa Cruz | Actin-HRP sc47778 | 1/5000 |
| Antibody | Tubulin (Rabbit polyclonal) | Abcam | ab15568 | 1/1000 |
| Antibody | Nup98 (Rat monoclonal) | Abcam | ab50610 | 1/1000 |
| Antibody | Histone H3 (Rabbit polyclonal) | Abcam | ab1791 | 1/1000 |
| Sequence-based reagent | Trp53 (c) | This paper | Primer for genotyping | GTGTGTTGGCCATCTCTGTG; *Figure 1* |

*Continued on next page*

*Continued*

| Reagent type (species) or resource | Designation | Source or reference | Identifiers | Additional information |
|---|---|---|---|---|
| Sequence-based reagent | Trp53 (d) | This paper | Primer for genotyping | AACCGGACTCAGCGTCTCTA; *Figure 1* |
| Sequence-based reagent | Trp53-F | *Fajac et al., 2024* | qPCR primer | AAAGGATGCCCATGCTACAGA; *Figure 1* |
| Sequence-based reagent | Trp53-R | *Fajac et al., 2024* | qPCR primer | TCTTGGTCTTCAGGTAGCTGGAG; *Figure 1* |
| Sequence-based reagent | Ccl17-F | This paper | qPCR primer | GCTGGTATAAGACCTCAGTGGAGTGT; *Figure 5* |
| Sequence-based reagent | Ccl17-R | This paper | qPCR primer | GCTTGCCCTGGACAGTCAGA; *Figure 5* |
| Sequence-based reagent | Ccl9-F | This paper | qPCR primer | GCACAGCAAGGGCTTGAAA; *Figure 5* |
| Sequence-based reagent | Ccl9-R | This paper | qPCR primer | CAGGCAGCAATCTGAAGAGTCTT; *Figure 5* |
| Sequence-based reagent | S100a8-F | This paper | qPCR primer | TCCTTTGTCAGCTCCGTCTTC; *Figure 5* |
| Sequence-based reagent | S100a8-R | This paper | qPCR primer | GACGGCATTGTCACGAAAGAT; *Figure 5* |
| Peptide, recombinant protein | Superscript IV | Invitrogen | TF #18090010 | |
| Commercial assay or kit | Nucleospin RNA II | Macherey-Nagel | FS #NZ74095520 | |
| Commercial assay or kit | Power SYBR Green | Applied Biosystems | #4367659 | |
| Commercial assay or kit | Supersignal West Femto | Thermo Fisher | #34096 | |
| Commercial assay or kit | Annexin V-FITC apoptosis staining/ detection kit | Abcam | #Ab14085 | |
| Commercial assay or kit | TruSeq Stranded Total RNA | Illumina | #20020596 | |
| Chemical compound, drug | Nutlin 3a | Sigma-Aldrich | #SML-0580 | |
| Chemical compound, drug | Doxorubicin | Sigma-Aldrich | #D1515 | |
| Chemical compound, drug | Supformin | *Solier et al., 2023* | | |
| Software, algorithm | FlowJo | Beckton-Dickinson | RRID:SCR_008520 | 10.10 |
| Software, algorithm | featureCounts | *Liao et al., 2014* | | |
| Software, algorithm | DESeq2 R package | *Love et al., 2014* | | |
| Software, algorithm | GOrilla | *Eden et al., 2009* | | |
| Software, algorithm | GSEA software | *Subramanian et al., 2005* | | |
| Software, algorithm | Prism | GraphPad | RRID:SCR_002798 | 5.0 |

## LSL-Y217C construct

We used mouse genomic *Trp53* DNA from previous constructs, containing a portion of intron 1 with a LoxP-Stop-LoxP (LSL) cassette (*Ventura et al., 2007*) in which the puromycin resistance gene was replaced by a neomycin resistance gene (*Simeonova et al., 2013*). The point mutation (A to G)

encoding a Tyr to Cys substitution (TAT to TGT) at codon 217, together with a silent mutation nearby creating a Ban II restriction site, were introduced by PCR directed mutagenesis by using primers Y217C-F and Y217C-R (see *Supplementary file 5* for primer sequences). The resulting *Trp53*[LSL-Y217C] targeting vector was fully sequenced before use.

## Targeting in ES cells and genotyping

129/SvJ ES cells were electroporated with the targeting construct linearized with Not I. Two recombinant clones, identified by long-range PCR and confirmed by internal PCR and Southern blot, were injected into C57BL/6J blastocysts to generate chimeras, and germline transmission was verified by genotyping their offspring. In vivo excision of the LSL cassette was performed by breeding *Trp53*[+/LSL-Y217C] male mice with females carrying the PGK-*Cre* transgene (*Lallemand et al., 1998*) and genotyping their offspring. The *Trp53*[Y217C] (p53[YC]) mutation was routinely genotyped by PCR using primers c and d (*Figure 1*, *Supplementary file 5*) followed by BanII digestion. Tumorigenesis studies were performed on mouse cohorts resulting from ≥5 generations of backcrosses with C57BL6/J mice. For all experiments, mice housing and treatment were conducted according to the Institutional Animal Care and Use Committee of the Institut Curie (approved project #03769.02).

## Cells and cell culture reagents

MEFs isolated from 13.5-day embryos were cultured in a 5% $CO_2$ and 3% $O_2$ incubator, in DMEM GlutaMAX (Gibco), with 15% FBS (Biowest), 100 μM 2-mercaptoethanol (Millipore), 0.01 mM Non-Essential Amino-Acids, and penicillin/streptomycin (Gibco) for ≤6 passages. Cells were irradiated with a Cs γ-irradiator or treated for 24 hr with 10 μM Nutlin 3a (Sigma-Aldrich) or 1 μM Doxorubicin (Sigma-Aldrich). Neurospheres were prepared by isolating neural stem/progenitor cells (NPCs) from the lateral and median ganglionic eminences of E14.5 embryos. NPCs were cultured in a 5% $CO_2$ incubator, in DMEM/F12 medium (with L-Glutamine, Invitrogen) supplemented with 1% B-27 (Invitrogen), 1% N2 (Invitrogen), 0.3% glucose (Invitrogen), 25 μg/ml insulin (Sigma-Aldrich), 20 ng/ml EGF (PeproTech), 10 ng/ml bFGF (PeproTech), and penicillin/streptomycin (Gibco). Neurospheres were dissociated once a week by Accutase treatment and mechanically, then seeded at $10^6$ cells per 6 cm diameter dish. Cell culture supernatants were tested and found negative for mycoplasma contamination.

## Quantitative RT-PCR

Total RNA, extracted using NucleoSpin RNA II (Macherey-Nagel), was reverse-transcribed using SuperScript IV (Invitrogen). Real-time quantitative PCRs were performed as previously described (*Simeonova et al., 2012*), on an ABI PRISM 7500 using Power SYBR Green (Applied Biosystems). Primer sequences are reported in *Supplementary file 5*.

## Western blots

Cells were lysed in RIPA buffer (50 mM Tris-HCl pH 8, 150 mM NaCl, 5 mM EDTA, 0.5% deoxycholic acid, 0.1% SDS, 1% NP-40) supplemented with Protease inhibitors cocktail (Roche) and 1 mM PMSF (Sigma). Whole-cell extracts were sonicated three times for 10 s and centrifuged at 13,000 rpm for 30 min to remove cell debris. Protein lysate concentration was determined by BCA assay (Thermo Scientific) and 30 μg of each lysate was fractionated by SDS-PAGE on a 4–12% polyacrylamide gel and transferred onto polyvinylidene difluoride membranes (Amersham). Membranes were incubated with antibodies raised against Mdm2 (MDM2-4B2, Abcam, 1/500), p53 (CM-5, Novocastra, 1/2000), p21 (F5, Santa Cruz Biotechnology, 1/200), and actin (actin-HRP sc47778, Santa Cruz Biotechnology, 1/5000). Chemiluminescence revelation of western blots was achieved with the SuperSignal West Dura (Perbio).

## ChIP assay

ChIP analysis was performed as previously described (*Simeonova et al., 2013*). p53-DNA complexes were immunoprecipitated from total extracts by using 50 μg of a polyclonal antibody against p53 (FL-393, Santa Cruz Biotechnology) and 300 μg of sonicated chromatin. Rabbit IgG (Abcam) was used for control precipitation. Quantitative PCR was performed on ABI PRISM 7500. Primer sequences are reported in *Supplementary file 5*.

## Cell fractionation assay

Cells were lysed in hypotonic lysis buffer (10 mM Tris pH 7.5, 10 mM NaCl, 3 mM MgCl$_2$, 0.3% NP-40, 10% glycerol) on ice for 10 min, then centrifuged at 4°C for 3 min at 1000×$g$. The supernatant was recovered as the cytoplasmic fraction, and the pellet was incubated in a modified Wuarin-Schibler buffer (10 mM Tris-HCl pH 7.0, 4 mM EDTA, 0.3 M NaCl, 1 M urea, 1% NP-40) on ice for 5 min, vortexed, incubated on ice for 10 min, then centrifuged at 4°C for 3 min at 1000×$g$. The supernatant was recovered as the nucleoplasmic fraction, and the pellet was washed, then incubated in a nuclear lysis buffer (20 mM Tris pH 7.5, 150 mM KCl, 3 mM MgCl$_2$, 0.3% NP-40, 10% glycerol) on ice for 5 min, sonicated, then centrifuged at 13,000 rpm at 4°C and the supernatant was recovered as the chromatin fraction. All buffers were supplemented with Protease inhibitors cocktail (Roche). Cellular fractions were analyzed by western blots with antibodies against p53 (AF-1355, R&D Systems, 1/600), Tubulin (ab15568, Abcam, 1/1000), Nup98 (ab50610, Abcam, 1/1000), and histone H3 (ab1791, Abcam, 1/1000). Chemiluminescence revelation was achieved with SuperSignal West Pico or Femto (Thermo Scientific) and analyzed with a ChemiDoc Imaging System (Bio-Rad). Scans for chemiluminescence were performed, with or without superposed colorimetric scans for molecular weights.

## Immunofluorescence

MEFs were cultured on collagen-coated coverslips, treated with 10 µM Nutlin 3a (Sigma-Aldrich) for 24 hr and analyzed. Coverslips were stained with rabbit anti-p53 FL-393 (Santa Cruz Biotechnology, 1/50), mouse anti-actin primary antibody and with Alexa Fluor 647 anti-Rabbit and Alexa Fluor 488 anti-Mouse secondary antibodies (Molecular Probes). DNA was counterstained with DAPI. Images were captured on an epifluorescence microscope using equal exposure times for all images for each fluor.

## Cell cycle assay

Log phase cells were irradiated at room temperature with a Cs γ-irradiator at doses of 3 or 12 Gy, incubated for 24 hr, then pulse labeled for 1 hr with BrdU (10 µM), fixed in 70% ethanol, double stained with FITC anti-BrdU and propidium iodide, and analyzed by flow cytometry with a BD Biosciences FACSort and the FlowJo software.

## Apoptosis assay

Six- to eight-week-old mice were left untreated or submitted to 10 Gy whole-body γ-irradiation. Mice were sacrificed 4 hr later and thymi were extracted. Thymocytes were recovered by filtration through a 70 µm smash, stained with Annexin V-FITC Apoptosis detection kit (Abcam) and propidium iodide, then analyzed with an LSRII FACS machine. Data were analyzed using FlowJo software.

## Metaphase spread preparation and analysis

Primary MEFs (at passage 4 for all genotypes) were treated with 0.1 mM nocodazole for 3 hr to arrest cells in metaphase, then submitted to hypotonic shock (75 mM KCl), fixed in a (3:1) ethanol/acetic acid solution, dropped onto glass slides, then air-dried slides were stained with Giemsa to score for chromosome aberrations. Images were acquired using a Zeiss Axiophot (×63) microscope.

## Histology

Organs were fixed in formol 4% for 24 hr, then ethanol 70%, and embedded in paraffin wax. Serial sections of 3 µm were stained as described (*Simeonova et al., 2013*), with hematoxylin and eosin using standard procedures.

## RNA-seq analysis

Total RNA was extracted from the thymi of 8-week-old asymptomatic male mice using nucleospin RNA II (Macherey-Nagel). The quality of RNA was checked with Bioanalyzer Agilent 2100 and RNAs with an RNA integrity number (RIN)>7 were retained for further analysis. RNA was depleted from ribosomal RNA, then converted into cDNA libraries using a TruSeq Stranded Total Library preparation kit (Illumina). Paired-end sequencing was performed on an Illumina MiSeq platform. Reads were mapped to the mouse genome version GRCm38 and counted on gene annotation gencode.vM18

with featureCounts (*Liao et al., 2014*). Differentially expressed genes with an adjusted p-value<0.05 were identified using the DESeq2 R package (*Love et al., 2014*).

## GO analysis

GO analysis of differentially expressed genes was performed by using the GOrilla (Technion) software as previously described (*Rakotopare et al., 2023*). Enrichment analyses were carried out by comparing the list of 192 differentially expressed genes between $Trp53^{YC/YC}$ and $Trp53^{-/-}$ thymi to the full list of genes (background), with ontology searches for biological processes and default p-value settings ($10^{-3}$).

## Gene set enrichment analysis

GSEA was performed by using the GSEA software with canonical pathway gene sets from the Mouse Molecular Signature Database (MSigDB) (*Subramanian et al., 2005*). Gene sets with a false discovery rate (FDR) q-value<0.25 were regarded as potentially relevant. The best candidates, with an FDR q-value<0.01, exhibited NES>1.91 and nominal p-values<0.003.

## Oral gavage of pregnant females

$Trp53^{+/YC}$ female mice were mated with $Trp53^{YC/YC}$ males, then mice with a vaginal plug on the next day were separated from males and weighted every other day to confirm pregnancy. On the 10th and 11th days post-coitum, pregnant mice received 5 mg/kg of supformin (in a PBS solution) by oral gavage with flexible gauge feeding tubes, then their pups were genotyped at weaning.

## Statistical analyses

Unless noted otherwise, we used a Student's t-test to analyze differences between two groups of values. For the proportions of mice at weaning, the observed mouse count was compared to the expected count according to Mendel's distribution and a chi-square ($\chi^2$) test, and a Fisher's test was applied to compare frequencies of mutant females or female to male ratios. Fisher's tests were also used to analyze frequencies of chromosome rearrangements in fibroblasts. Survival curves were analyzed with log-rank Mantel-Cox tests. Analyses were performed by using GraphPad Prism, and values of p≤0.05 were considered significant.

## Acknowledgements

This project was supported by grants attributed to FT, from the Ligue Nationale Contre le Cancer, the Fondation ARC pour la recherche sur le Cancer and the Gefluc. SJ, EE, and JR were PhD fellows of the Ministère de la Recherche, supervised by FT (SJ, JR) or BB (EE). SJ and EE received additional support from the Fondation ARC. MG was paid by European Research Council 875532-Prostator-ERC-2019-PoC attributed to AM. For some experiments, SJ and VV received technical assistance from C Bouyer and R Bourimi. We thank the following members of the Institut Curie platforms: I Grandjean, H Gautier, C Daviaud, M Garcia, D Andreau, M Verlhac, and A Fosse (animal facility); S Baulande and S Lameiras (NGS); M Huerre, A Nicolas, and R Leclere (histopathology); Z Maciorowski, A Viguier, and S Grondin (flow cytometry).

## Additional information

### Funding

| Funder | Grant reference number | Author |
|---|---|---|
| Fondation ARC pour la Recherche sur le Cancer | Projet | Franck Toledo |
| Ligue Contre le Cancer | Labellisation 14-18 | Franck Toledo |
| Groupement des Entreprises Françaises dans la lutte contre le Cancer | Projet Métastases | Franck Toledo |

| Funder | Grant reference number | Author |
|---|---|---|
| European Research Council | 10.3030/875532 | Antonin Morillon |

The funders had no role in study design, data collection and interpretation, or the decision to submit the work for publication.

## Author contributions
Sara Jaber, Eliana Eldawra, Jeanne Rakotopare, Iva Simeonova, Vincent Lejour, Vitalina Volochtchouk, Monika Licaj, Anne Fajac, Investigation; Marc Gabriel, Formal analysis; Tatiana Cañeque, Raphaël Rodriguez, Antonin Morillon, Resources; Boris Bardot, Formal analysis, Supervision, Investigation, Visualization, Writing – original draft; Franck Toledo, Conceptualization, Formal analysis, Supervision, Funding acquisition, Investigation, Visualization, Writing – original draft, Project administration, Writing – review and editing

## Author ORCIDs
Vincent Lejour https://orcid.org/0000-0002-2797-1507
Antonin Morillon https://orcid.org/0000-0002-0575-5264
Boris Bardot https://orcid.org/0000-0003-4976-9593
Franck Toledo https://orcid.org/0000-0003-3798-4106

## Ethics
For all experiments, mice housing and treatment were conducted according to the Institutional Animal Care and Use Committee of the Institut Curie (approved project #03769.02).

Reviewer #1 (Public review): https://doi.org/10.7554/eLife.102434.3.sa1
Reviewer #2 (Public review): https://doi.org/10.7554/eLife.102434.3.sa2
Author response https://doi.org/10.7554/eLife.102434.3.sa3

# Additional files

## Supplementary files
Supplementary file 1. Differentially expressed genes between wildtype (WT), *Trp53*^Y217C/Y217C^ (YC), and *Trp53*^-/-^ (KO) thymocytes from age-matched animals. Most differentially expressed genes correspond to a loss of function (LOF) in the p53^Y217C^ mutant (categories 1 and 2, see *Figure 4—figure supplement 3A*), but genes corresponding to a separation of function (SOF, categories 3 and 4) or a gain of function (GOF, categories 5 and 6) were also observed.

Supplementary file 2. A DESeq analysis of RNA-seq data from *Trp53*^-/-^ (KO) and *Trp53*^Y217C/Y217C^ (YC) thymi revealed 192 differentially expressed genes. As shown in *Figure 5B*, 107 genes (in bold) were less expressed and 85 genes were more expressed in *Trp53*^YC/YC^ cells compared to *Trp53*^-/-^ cells.

Supplementary file 3. *Ccl17*, *Ccl9*, *Ccr3*, *Cxcl10*, *S100a8,* and *S100a9* are associated with gene ontology (GO) terms of white blood cell migration/chemotaxis and inflammation.

Supplementary file 4. Distribution of weaned pups after mating *Trp53*^+/YC^ females with *Trp53*^YC/YC^ males and oral gavage of pregnant females with supformin. Due to the gavage of pregnant mothers, all the pups born in this experiment were exposed to supformin in utero (SIU+).

Supplementary file 5. Primers used in this study.

MDAR checklist

## Data availability
RNA sequencing data have been deposited in the Gene Expression Omnibus (GEO) under the accession code GSE248936. All other data are available within the article and its supplementary information. Materials can be provided by Franck Toledo pending scientific review and a completed material transfer agreement. Requests for materials should be submitted to franck.toledo@sorbonne-universite.fr.

The following dataset was generated:

| Author(s) | Year | Dataset title | Dataset URL | Database and Identifier |
|---|---|---|---|---|
| Bardot B, Toledo F, Gabriel M | 2025 | Thymocytes mRNA profiles of 8 weeks-old p53+/+, p53Y217C/Y217C and p53-/- male mice | https://www.ncbi.nlm.nih.gov/geo/query/acc.cgi?acc=GSE248936 | NCBI Gene Expression Omnibus, GSE248936 |

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
