## [Editor Report · eLife Assessment]

This work is of **fundamental** significance and has an **exceptional** level of evidence for the role of a mutant p53 in regulation of tumorigenesis using an in vivo mouse model. The study is well-conducted and will be of interest to a broad audience including those interested in p53, transcription factors and cancer biology.

---

## [Referee Report · Reviewer #1 (Public review)]

Summary:

This manuscript by Toledo and colleagues describes the generation and characterization of Y220C mice (Y217C in the mouse allele). The authors make notable findings: Y217C mice that have been backcrossed to C57Bl/6 for five generations show decreased female pup births due to exencephaly, a known defect in p53 -/- mice, and they show a correlation with decreased Xist expression, as well as increased female neonatal death. They also noted similar tumor formation in Y217C/+ and p53 +/- mice, suggesting that Y217C may not function as a dominant negative. Notably, the authors find that homozygous Y217C mice die faster than p53 -/- mice, and that the lymphomas in the Y217C mice were more aggressive and invasive. The authors then perform RNA seq on thymi of Y217C homozygotes compared to p53 -/-, and they suggest that these differentially expressed genes may explain the increased tumorigenesis in Y217C mice.

Strengths:

Overall, the study is well controlled and quite well done and will be of interest to a broad audience, particularly given the high frequency of the Y220C mutation in cancer (1% of all cancers, 4% of ovarian cancer).

Weaknesses:

None noted

Comments on revisions:

The authors have done a superb job on this very interesting work.

---

## [Referee Report · Reviewer #2 (Public review)]

Summary:

Jaber et al. describe the generation and characterization of a knock-in mouse strain expressing the p53 Y217C hot-spot mutation. While the homozygous mutant cells and mice reflect the typical loss-of-p53 functions, as expected, the Y217C mutation also appears to display gain-of function (GOF) properties, exemplified by elevated metastasis in the homozygous context (as noted with several hot-spot mutations). Interestingly, this mutation does not appear to exhibit any dominant-negative effects associated with most hot-spot p53 mutations, as determined by absence of differences in overall survival and tumor predisposition of the heterozygous mice, as well as target gene activation upon nutlin treatment.

In addition, the authors noted a severe reduction in the female 217/217 homozygous progeny, significantly more than that observed with the p53 null mice, due to exencephaly, leading them to conclude that the Y217C mutation also has additional, non-cancer related GOFs. Thought this property has been well described and attributed to p53 functional impairment, the authors conclude that the Y217C has additional properties in accelerating the phenotype.

Transcriptomic analyses of thymi found additional gene signature differences between p53 null and the Y217C strains, indicative of novel target gene activation, associated with inflammation.

Strengths:

Overall, the characterisation of the mice highlights the expected typical outcomes associated with most hot-spot p53 mutations published earlier. The quality of the work presented is well done and good, and the conclusions and reasonably well justified.

Comments on revisions:

Revised version has addressed most of our queries and is acceptable.

---

## [Author Response]

The following is the authors’ response to the original reviews

Reviewer #2 suggested the addition of new data to address the following points:

**Reviewer #2:**
(1) Oncogenic GOF - the main data shown for GOF are the survival curve and enhanced metastasis. Often, GOF is exemplified at the cellular level as enhanced migration and invasion, which are standard assays to support the GOF. As such, the authors should perform these assays using either tumor cells derived from the mice or transformed fibroblasts from these mice. This will provide important and confirmatory evidence for GOF for Y217C.

We thank the referee for this comment. Our previous data indicated accelerated tumor progression and increased metastasis in *Trp53*^Y217C/Y217C^ mice, which provided in vivo evidence of an oncogenic gain of function (GOF) for the p53^Y217C^ mutant. However, we agree that it was important to provide additional evidence of GOF at the cellular level.

Many cellular assays were previously used to evaluate the GOF of p53 mutants, including those listed by the referee. Importantly, Zhao et al. recently showed that a common property of several p53 mutants proposed to have oncogenic GOF is their capacity to promote chromosomal instability (Zhao et al. (2024) Nat. Commun. 15, 180). For the revision of our manuscript, we compared the frequencies of chromosomal alterations occurring spontaneously in WT, *Trp53*^Y217C/Y217C^ and *Trp53-/-* mouse embryonic fibroblasts (MEFs). Chromosome breaks, radial chromosomes and DMs were more frequent in *Trp53*^Y217C/Y217C^ MEFs than in WT or *Trp53-/-* MEFs, providing clear evidence of a GOF promoting chromosomal instability. This new result is presented in Figure 2G and mentioned in the revised abstract.

Furthermore, as pointed out by referee #1 in a confidential comment, increased NF-kB signaling provides evidence of p53 GOF. Accordingly, Zhao et al. proposed that the capacity of p53^G245D^ and p53^R273H^ to promote chromosomal instability ultimately led to activation of a noncanonical NF-kB signaling that would promote tumor cell invasion and metastasis. Consistent with their work, we now report that the GSEA of *Trp53*^Y217C/Y217C^ and *Trp53-/-* thymocytes revealed an upregulation of non-canonical NF-kB signaling in *Trp53*^Y217C/Y217C^ thymic cells (a new result presented in Figure 5F and Supplementary Figure S13). These new data lead us to mention in the revised discussion that “similar mechanisms might underlie the oncogenic properties of the p53^Y217C^, p53^G245D^ and p53^R273H^ mutants”.

(2) Novel target gene activation - while a set of novel targets appears to be increased in the Y217C cells compared to the p53 null cells, it is unclear how they are induced. The authors should examine if mutant p53 can bind to their promoters through CHIP assays, and, if these targets are specific to Y217C and not the other hot-spot mutations. This will strengthen the validity of the Y217C's ability to promote GOF.

We respectfully disagree with the referee when he/she considers that the validity of p53^Y217C^’s ability to promote a GOF would be strengthened by showing that p53^Y217C^ binds to the promoters of genes upregulated in *Trp53*^Y217C/Y217C^ cells. In fact, Pal *et al.* recently performed the experiment proposed by the referee, by integrating RNAseq and ChIPseq data from MCF10A cells expressing p53^Y220C^, the human equivalent of p53^Y217C^, and found that 95% of the genes upregulated upon p53^Y220C^ expression were upregulated indirectly, without p53^Y220C^ binding to their promoters (Pal et al. (2023) NPJ Breast Cancer 9, 78). Consistent with our data, Pal et al. notably found that the expression of p53^Y220C^ increased cell migration and invasion, which correlated with an increased expression of S100A8 and S100A9. They found that the promoters of S100A8 and S100A9 were however not bound by p53^Y220C^, indicating an indirect mechanism for their upregulated expression. Furthermore, the study by Zhao et al. mentioned above also suggested an indirect mechanism of GOF, because the upregulation of inflammation-related genes by a mutant p53 protein was proposed to result from signaling cascades triggered by chromosomal instability. Our data appear consistent with both studies, because p53^Y217C^ was undetectable or barely detectable in the chromatin fraction of *Trp53*^Y217C/Y217C^ cells, and because *Trp53*^Y217C/Y217C^ cells exhibited increased chromosome instability and increased NFκB signaling compared to *Trp53-/-* cells, which may suggest indirect mechanisms for p53^Y217C^ GOF.

Nevertheless, we agree with the referee that it was important to provide stronger evidence of p53^Y217C^ GOF in the revised manuscript. In that regard, we were intrigued by the perinatal death of most *Trp53*^Y217C/Y217C^ females, which provided evidence of unexpected teratogenic effects of the mutant. We had proposed that these female-specific teratogenic effects likely resulted from pro-inflammatory GOF of p53^Y217C^. This hypothesis relied on the RNAseq pro-inflammatory signature in *Trp53*^Y217C/Y217C^ thymic cells, and on the fact that the glycoprotein CD44, known to drive inflammation, had been identified as a key gene in open neural tube defects. However, we had not tested this hypothesis experimentally. In the revised version of the manuscript, we tested this hypothesis. We mated *Trp53*^+/Y217C^ female mice with *Trp53*^Y217C/Y217C^ males, then administered supformin (LCC-12), a potent CD44 inhibitor known to attenuate inflammation in vivo, to pregnant mice by oral gavage. The administration of subformin led to a five-fold increase in the proportion of weaned *Trp53*^Y217C/Y217C^ females in the progeny, suggesting that reducing inflammation in utero rescued some of the *Trp53*^Y217C/Y217C^ female embryos. This new result is presented in Figure 5G and Supplementary Table S6, and mentioned in the abstract.

We believe that these new results, as well as the additional GSEA analyses revealing increased NFkB signaling in *Trp53*^Y217C/Y217C^ cells, further emphasize the importance of inflammation in the GOF of the p53^Y217C^ mutant. Accordingly, we slightly modified the title of our article, to include the notion that *Trp53*^Y217C^ is an inflammation-prone mouse model. We also end the article by summarizing the effects of p53^Y217C^ in vivo, in a new Supplementary Table S7 that compares the LOF effects of a p53 KO with the (LOF+GOF) effects of the p53^Y217C^ mutant.

(3) Dominant negative effect - the authors' claim of lack of DN effect needs to be strengthened further, as most p53 hot-spot mutations do exhibit DN effect. At the minimum, the authors should perform additional treatment with nutlin and gamma irradiation (or cytotoxic/damaging agents) and examine a set of canonical p53 target genes by qRT-PCR to strengthen their claim.

Our previous data indicated identical tumor onset and survival in *Trp53*^+/Y217C^ and *Trp53*^+/-^ mice, leading us to conclude that, at least for spontaneous tumorigenesis, there was no evidence of a Dominant Negative Effect (DNE) in vivo. Here, we followed the referee’s suggestion and evaluated the possibility of a DNE in response to stress, by comparing WT, *Trp53*^+/Y217C^ and *Trp53*^+/-^ MEFs or thymocytes. We analyzed different types of stress (Nutlin, Doxorubicin, girradiation) and different types of cellular responses (transactivation of classical p53 target genes, cell cycle arrest, apoptosis), and the results lead us to conclude that there is little if any DNE also in response to various stresses. These new data are mentioned in a paragraph evaluating the possibility of DNE or GOF at the cellular level, and presented in a new Supplementary Figure S6.